evolution

postcopulatory sexual selection, sperm design, sperm heteromorphism, fertilization

**Author for correspondence:**
John L. Fitzpatrick
e-mail: john.fitzpatrick@zoologi.su.se

# Repeated evidence that the accelerated evolution of sperm is associated with their fertilization function

John L. Fitzpatrick[1,2], C. Daisy Bridge[2] and Rhonda R. Snook[1]

[1]Department of Zoology, Stockholm University, Svante Arrhenius väg 18B, SE-10691, Sweden
[2]Faculty of Life Sciences, University of Manchester, Michael Smith Building, Manchester M13 9PT, UK

JLF, 0000-0002-2834-4409

Spermatozoa are the most morphologically diverse cell type, leading to the widespread assumption that they evolve rapidly. However, there is no direct evidence that sperm evolve faster than other male traits. Such a test requires comparing male traits that operate in the same selective environment, ideally produced from the same tissue, yet vary in function. Here, we examine rates of phenotypic evolution in sperm morphology using two insect groups where males produce fertile and non-fertile sperm types (*Drosophila* species from the *obscura* group and a subset of Lepidoptera species), where these constraints are solved. Moreover, in *Drosophila* we test the relationship between rates of sperm evolution and the link with the putative selective pressures of fertilization function and postcopulatory sexual selection exerted by female reproductive organs. We find repeated evolutionary patterns across these insect groups—lengths of fertile sperm evolve faster than non-fertile sperm. In *Drosophila*, fertile sperm length evolved faster than body size, but at the same rate as female reproductive organ length. We also compare rates of evolution of different sperm components, showing that head length evolves faster in fertile sperm while flagellum length evolves faster in non-fertile sperm. Our study provides direct evidence that sperm length evolves more rapidly in fertile sperm, probably because of their functional role in securing male fertility and in response to selection imposed by female reproductive organs.

## 1. Introduction

Sperm are the most diverse cell type despite their homologous function across taxa of securing male fertility [1,2]. This diversity suggests sperm evolve rapidly, which is commonly hypothesized to be due to selection for their fertilization function and in response to the postcopulatory processes of sperm competition and selection imposed by the female reproductive tract (i.e. cryptic female choice [1–5]). Postcopulatory sexual selection in particular is often credited as a main driver of phenotypic diversification [6–8]. Indeed, numerous studies now demonstrate that sexually selected traits evolve faster than naturally selected ecological and life-history traits (e.g. [6–10]) and phenotypic diversification of sexual traits are exaggerated in species where the strength of sexual selection is stronger (e.g. [11–13]). However, there is no direct evidence that sperm morphology shows an accelerated rate of evolution or that sperm evolve faster than other male traits, including other sexually selected traits.

Three studies have attempted to examine rates of sperm morphological evolution. In *Onthophagine* dung beetles and *Anolis* lizards, sperm length showed a *slower* rate of phenotypic diversification than other precopulatory (i.e. horn length), postcopulatory (i.e. testes size) or somatic (i.e. body size) male traits [9,14]. However, in these studies, rates of sperm diversification were compared with traits that operate in different selective environments—traits located outside (e.g. horns) or inside (e.g. testes) the body—and during different episodes of

sexual selection (i.e. precopulatory versus postcopulatory). This makes it challenging to draw general conclusions about the causes of different rates of evolution between sperm and other phenotypic traits as the strength of, and constraints on, selection may be context dependent and different traits likely have different developmental and physical constraints influencing their diversification. The third study found that sperm midpiece and flagellum length evolved *faster* in bird lineages where males invested more in testes size (a proxy for sperm competition risk), suggesting that postcopulatory sexual selection promotes elevated rates of sperm phenotypic diversification [12]. However, whether sperm evolve faster or slower than other sexually selected traits in birds remains unclear, as sperm length was not compared against other phenotypic traits [12]. A robust test of the hypothesis that sperm evolve faster than other traits requires a model where the male traits under study operate in the same environment, ideally produced from the same tissue, yet vary in their function.

Sperm consist of different components (i.e. head, midpiece (in some species) and flagellum) that work as a functional unit to ensure fertilization. Thus, selection on any component may be constrained via trade-offs or genetic correlations by virtue of the joint impact of different components on the ability of sperm to secure fertilizations [15,16]. Relatively few studies have quantified the genetic variation and covariation of sperm components, but the available data suggest that genetic correlations are prevalent [15]. However, the direction of these relationships varies. For example, sperm midpiece and flagellum length exhibit negative genetic correlations in zebra finches, *Taeniopygia guttata* [17], whereas sperm head (there is no midpiece) and flagellum length exhibit positive genetic correlations in *Drosophila pseudoobscura* [16]. Regardless of the direction, such correlations imply that suites of sperm traits, rather than individual sperm traits in isolation, will be selected. The extent to which different sperm components show different evolutionary rates has only been tested previously in passerine birds; head length showed a different evolutionary trajectory to midpiece and flagellum length [12,18]. Thus, even though sperm may be regarded as an integrated, and selected unit, different components may evolve independently.

Here we examine the rates of phenotypic evolution in sperm morphology in *Drosophila* from the *obscura* group and Lepidoptera (moths and butterflies). These two groups exhibit sperm heteromorphism, where males produce two sperm morphs within a single ejaculate that are either fertilization competent or non-fertile (i.e. sperm that are either directly or not/indirectly involved in fertilization, respectively, [19,20]). In the *obscura* group males produce two types of nucleated sperm, but only the long *eusperm* morph participates in fertilization while the short *parasperm* morph is non-fertile and never found inside eggs [19,20]. Lepidoptera produce long, nucleated *eupyrene* sperm that are fertilization competent, and short, non-fertile *apyrene* sperm that lack nuclear material [19,20]. In both of these insect groups, fertile and non-fertile sperm are produced in different cysts within the same testes and are transferred together, and stored in, the female reproductive tract upon mating [19,20]. This provides a rare opportunity to contrast the rate of phenotypic evolution in sperm morphology between cells that are produced from the same tissue and operate in the same environment, but where only one morph plays a direct role in fertilization. Non-fertile sperm serve different adaptive functions in the few species that have been studied [21]. Non-fertile sperm can protect fertile sperm from a hostile female

reproductive tract (e.g. *D. pseudoobscura* [22]), facilitate fertile sperm migration in the female reproductive tract (e.g. *Bombyx mori* [23]) and/or act as 'cheap fillers' of the female sperm storage organ, thereby reducing sperm competition by delaying female remating rates (e.g. *Pieris napi* [24]). Comparative studies reveal that dimensions of female reproductive organs positively relate to fertile sperm length but exhibit no relationship with non-fertile sperm length [25,26]. Sperm competition risk influences each sperm type differently between taxa [26,27]. For fertile sperm length, Lepidoptera show a positive relationship with sperm competition risk but there is no relationship between these traits in *obscura* group species [25–27]. For non-fertile sperm length, flies show a negative relationship with sperm competition risk, while Lepidoptera show mixed patterns [25–27]. Thus, there are substantial differences in how selection operates on fertile and non-fertile sperm length, providing an opportunity to contrast the evolution of sperm morphology between these two groups.

Here, we test the prediction that fertile and non-fertile sperm length evolve at different rates given their different functions by directly quantifying and comparing the evolutionary rates between sperm morphs. We also contrast the rates of phenotypic evolution in fertile and non-fertile sperm length with rates of evolution in the female reproductive organ length (in *Drosophila*) and with male wing length, a proxy for male body size (in *Drosophila* and Lepidoptera). Finally, as longer sperm are often competitively superior in sperm monomorphic species (e.g. *D. melanogaster* [28]), we decompose how postcopulatory sexual selection and fertilization may influence evolutionary rates on different sperm components, testing the prediction that flagellum length will exhibit faster evolutionary rates than head length in fertile sperm.

## 2. Methods

### (a) Data collection

Data on fertile and non-fertile sperm length, somatic traits and female reproductive organ length were compiled from the literature for *Drosophila* from the *obscura* group and Lepidoptera (electronic supplementary material, table S1). For *Drosophila*, data on fertile and non-fertile sperm length were collected from 19 species from a range of sources. Male wing length, which was used as a proxy measure for male body size given its tight correlation with male thorax mass [29], was available for 18 of these species, and seminal receptacle length, a female reproductive tract organ that is the primary sperm storage organ for sperm that are first used for fertilization [30], was available for 15 of these species. Lepidoptera data came from two sources [26,27] and included data on fertile and non-fertile sperm length and male forewing length, which is tightly correlated with male body mass [27], for 135 species. Only 12 of which were present in the phylogeny used for the analyses. Data on female reproductive organs were available for a subset of species [26], only one of which was present in the phylogeny used for the analyses (see below). Therefore, we did not assess female reproductive organs in the Lepidoptera analyses. Given their primary function in locomotion, male wing/forewing length is less likely to be influenced exclusively by sexual selection and therefore we treat wing/forewing length as a point of comparison against sperm and female reproductive organ length.

### (b) *Drosophila* and Lepidoptera Phylogenies

We constructed a phylogeny for *Drosophila* from the *obscura* group using available sequences (see below). For Lepidoptera,

we used a recent phylogeny of 186 species constructed using a large genomic and transcriptomic sequence dataset [31].

*Drosophila* from the *obscura* group are a monophyletic group that arose approximately 15–20 million years ago [32,33]. We used Geneious (8.0.2, Biomatters Ltd) to search GenBank for candidate genes for species where data were available on sperm morphology (electronic supplementary material, table S1). We constructed a phylogeny using the mitochondrial gene cytochrome c oxidase subunit II (COII). The high substitution rate of mitochondrial gene makes COII useful for estimating divergence times among the closely related and relatively recently diverging *obscura* species. Nucleotides sequences were aligned in Mesquite version 3.40 [34] using the MUSCLE extension for this program [35] and inspected to identify problematic alignments. We used jModelTest v2.1.6 [36] to identify the best-fit nucleotide substitution model for each sequence. The Akaike information criterion corrected for sample size (AICc) was used to distinguish between three substitution models. The best-fit nucleotide substitution models were GTR + $\Gamma$. However, to facilitate model convergence during the tree-building stage, we simplified the substitution models to GTR, which led to convergence of the Bayesian chain.

Phylogenies were constructed using the Bayesian tree-building analysis software BEAUTi v1.8.2 and BEAST v1.8.2 [37], using a relaxed uncorrelated lognormal clock, default BEAST values for sequence-evolution parameters, a Yule speciation process and no priors set for root dates. Priors were set to a uniform distribution with an initial value of 0.6 and upper and lower values of 1 and 0, respectively. For the *Drosophila* phylogeny, we assigned species to five previously identified monophyletic subgroups within the *obscura* group (the *affinis*, *microlabis*, *subobscura*, *pseudoobscrua* and *obscura* subgroups; [38]). The Markov chain Monte Carlo (MCMC) simulation chain generated by BEAST was set to a length of 10 million steps with parameters logged every 10 000 steps. Stationarity was verified using Tracer v1.5 software [39], based on inspection of the posterior distribution of the traces and an effective sample size (ESS) exceeding 200 for each parameter. A maximum clade credibility (MCC) tree was generated using mean node heights and a 10% burn-in, and a posterior probability limit of 0.90 using TreeAnnotator v1.8.1 [37], and viewed using FigTree v1.4.2 [37]. To verify the resulting phylogeny, we re-ran the steps described above using alternative staring points and a greater number of sequences for *Drosophila* from *obscura* group (COII, 28S ribosomal RNA gene and cytochrome b). In all cases, the topology remained entirely or largely consistent regardless of how the phylogenies were generated. Our fully resolved phylogeny for *Drosophila* from the *obscura* group (electronic supplementary material, figure S1) is consistent with previously published molecular phylogenies from this group [33,38].

## (c) Phylogenetic analyses

Using the time-calibrated molecular phylogenies for each group (electronic supplementary material, figures S1 and S2), we compared the rates of evolution between fertile and non-fertile sperm length, male wing/forewing length and female reproductive organ length (for *Drosophila*) using Adams's [40] likelihood-based approach (see electronic supplementary material, for additional details). We evaluated and compared the evolutionary Brownian rate parameter, $\sigma^2$, which describes the rate of trait diversification across a phylogeny, of multiple traits [40]. For each analysis, the observed evolutionary rate matrix ($\sigma^2_{obs}$) was determined for each trait and contrasted against a constrained model that assumes all traits evolve at a common evolutionary rate ($\sigma^2_{common}$). Likelihood ratio tests were used to compare the observed and common evolutionary models.

# 3. Results

## (a) Comparing rates of evolution in *Drosophila* from the obscura group

Across the *obscura* group, analyses comparing rates of evolution among sperm length, male wing length and seminal receptacle length revealed that fertile sperm length evolves significantly faster, at 3.4 times the rate, than non-fertile sperm length (table 1*a* and figure 1*a*). Fertile sperm length also evolved at a similar evolutionary rate as seminal receptacle length (i.e. the length of the female sperm storage organ), while both fertile sperm length and seminal receptacle length evolve faster than non-fertile sperm length (table 1*a*, figure 1*a*; electronic supplementary material, table S4). Male wing length evolved significantly slower than all other phenotypic traits examined (table 1*a*, figure 1*a*, electronic supplementary material, table S4).

The components of fertile and non-fertile sperm evolved at different evolutionary rates (table 2). Head length evolved almost twice as fast as flagellum length in fertile sperm (table 2*a*), whereas flagellum length evolved over four times as fast as head length in non-fertile sperm (table 2*b*). The reduced rate of evolution in non-fertile sperm head length was evident when we compared the rate of evolution between fertile and non-fertile sperm head length, an analysis that revealed that fertile sperm head length evolved 21.5 times as fast as non-fertile sperm head length (table 2*c*). There was a statistical trend ($p = 0.06$) suggesting that flagellum length of fertile sperm evolved faster than that of non-fertile sperm (table 2*d*).

Since our analyses revealed that evolutionary rates differed significantly within and between sperm morphs, we performed a final set of analyses comparing the rate of evolutionary diversification in seminal receptacle length with that of sperm head and flagellum length for each sperm morph. Seminal receptacle length evolved at a statistically indistinguishable rate from both fertile sperm head and flagellum length (table 2*e*,*f*). By contrast, seminal receptacle length evolved significantly faster than both non-fertile sperm head length and flagellum length (table 2*f*,*g*).

## (b) Comparing rates of evolution in Lepidoptera

Among Lepidoptera, fertile sperm length evolved 3.8 times faster than non-fertile sperm length, while the rate of evolution in forewing length was statistically indistinguishable from either sperm type (table 1*b*, figure 1*b*; electronic supplementary material, table S4). We found the same pattern for sperm length evolution when we tested against a phylogeny with more species ($n = 135$) but that was generated from a single sequence (see electronic supplementary material, Lepidoptera Analyses).

# 4. Discussion

We provide consistent evidence that fertile sperm length evolves faster than non-fertile sperm length in *Drosophila* from the *obscura* group and in a group of Lepidoptera for which sperm data were available, two groups with heteromorphic sperm. This repeated pattern was observed despite substantial physiological and adaptive differences in sperm heteromorphism among taxa. While previous studies have examined evolutionary rate of sperm, these studies compared sperm rate evolution to traits functioning in other episodes of

**Table 1.** Comparisons of evolutionary rates assuming trait covariance in the observed rate matrices of sperm length, female reproductive organ length and male wing length in *Drosophila* from the *obscura* group (*a*) and Lepidoptera (*b*). Separate models were performed comparing evolutionary rates of fertile (eusperm or eupyrene) and non-fertile (parasperm or apyrene) sperm total length ($L_{total}$) for each insect group, including female reproductive organ length and male wing length. For female reproductive organs, seminal receptacle length was examined in *Drosophila* from the *obscura* group and spermathecal duct length was examined for moths. The sample size (*n*), observed ($\sigma^2_{obs}$) and common ($\sigma^2_{common}$) rate matrices are shown for each analysis. The evolutionary rates in the diagonal of the rate matrix is presented in bold as these illustrate difference or similarities in the observed evolutionary rates among traits and how they differ from the common evolutionary rate (presented in brackets). The log-likelihood values for the observed ($LogL_{obs}$) and common models ($LogL_{common}$), log-likelihood ratio tests (LRT), with associated degrees of freedom (df), comparing models of observed rates with evolutionarily constrained models where all traits evolve at a common rate, p-values and AIC values for the observed ($AIC_{obs}$) and common ($AIC_{common}$) models are presented. Significant p-values are presented in italic text.

| trait | n | $\sigma^2_{obs}$ ($\sigma^2_{common}$) | $\sigma^2_{obs}$ ($\sigma^2_{common}$) | $\sigma^2_{obs}$ ($\sigma^2_{common}$) | $\sigma^2_{obs}$ ($\sigma^2_{common}$) | $LogL_{obs}$ ($LogL_{common}$) | LRT (df) | p | $AIC_{obs}$ ($AIC_{common}$) |
|---|---|---|---|---|---|---|---|---|---|
| **(a) Drosophila** | | fertile $L_{total}$ | non-fertile $L_{total}$ | wing length | seminal receptacle | | | | |
| fertile $L_{total}$ | 15 | *0.99 (0.62)* | — | — | — | 46.71 (25.54) | 42.32 (df = 3) | *<0.001* | −77.41 (−41.09) |
| non-fertile $L_{total}$ | | 0.24 (0.26) | *0.29 (0.62)* | — | — | | | | |
| wing length | | 0.09 (0.17) | 0.02 (0.47) | *0.03 (0.62)* | — | | | | |
| seminal receptacle | | 0.90 (0.49) | 0.09 (0.02) | 0.06 (0.002) | *1.09 (0.62)* | | | | |
| **(b) Lepidoptera** | | fertile $L_{total}$ | non-fertile $L_{total}$ | forewing length | | | | | |
| Fertile $L_{total}$ | 12 | *0.053 (0.034)* | — | — | | 21.35 (17.10) | 8.50 (df = 2) | *0.01* | −30.69 (−26.19) |
| non-fertile $L_{total}$ | | 0.015 (0.02) | *0.014 (0.034)* | — | | | | | |
| forewing length | | 0.014 (0.01) | 0.012 (0.02) | *0.032 (0.034)* | | | | | |

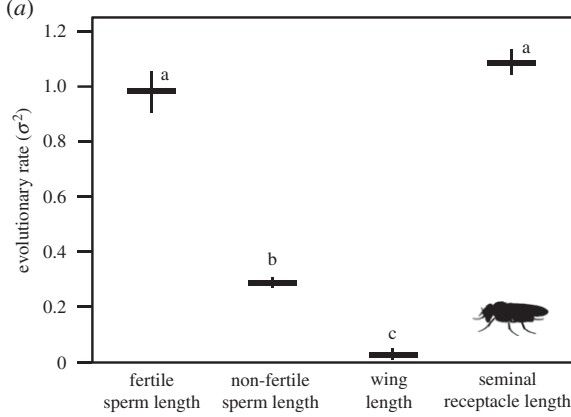

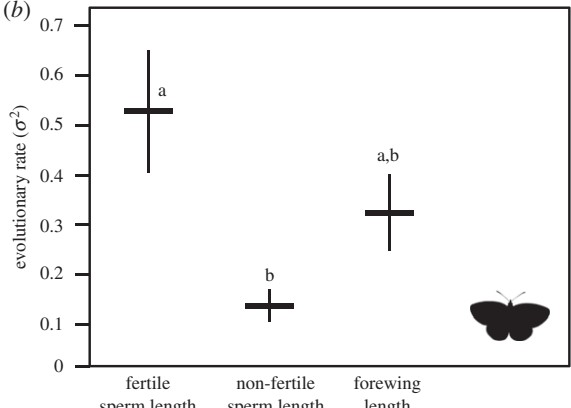

**Figure 1.** Estimates of the observed evolutionary rates ($\sigma^2$) of phenotypic evolution, with 95% confidence intervals, for log10-transformed values of reproductive and somatic traits in (*a*) *Drosophila* from the *obscura* group and (*b*) Lepidoptera. Different letters denote significant differences in rates of phenotypic evolution among traits based on post hoc pairwise comparisons (see electronic supplementary material, table S4). Note that $\sigma^2$ values are only comparable among traits from the same analyses and are therefore not directly comparable between *Drosophila* and Lepidotera. Silhouettes were downloaded from http://www.phylopic.org and are licensed for use in the public domain without copyright.

sexual selection (e.g. precopulatory versus postcopulatory episodes [9,14]). Contrary to the prediction of rapid sperm diversification, these studies found that sperm morphology evolved more slowly than non-sperm traits [9,14]. Our analysis is the first that examines rates of sperm diversification when traits operate in the same selective environment—the female reproductive tract. We do this by focusing on species where males produce fertile and non-fertile sperm morphs and find robust support that the fertilization function of sperm drives its rapid evolutionary diversification. Moreover, like other sexually selected traits [8], sperm are expected to exhibit accelerated rates of phenotypic diversification relative to naturally selected traits. In support of this hypothesis, we found that fertile sperm length evolves faster than body size in *Drosophila*, although this was less clear in Lepidoptera, likely due to reduced sample size in our analyses. Together, our results support the idea that the tight link with fertilization drives the diversification of sperm phenotypes observed across animals.

The rate of fertile sperm length diversification was equivalent to the rate of seminal receptacle length diversification in *Drosophila*. The similarity in the rates of phenotypic diversification between female reproductive organ length and sperm

length highlights the importance of considering female sperm co-evolutionary dynamics in internally fertilizing species. Indeed, fertile sperm size is positively associated with dimensions of female reproductive organs in both *Drosophila* from the *obscura* group and moths [25,26], while in sperm monomorphic taxa, there are numerous interspecific demonstrations of phenotypic correlations between female reproductive organ dimensions and sperm morphology (e.g. [30]). Moreover, experimental work demonstrates that the female reproductive tract is the selective agent driving rapid changes in sperm length in sperm monomorphic taxa [41]. If, as seems to be the case, female reproductive organs are a common selective force driving phenotypic diversification in sperm length among internal fertilizers, then the tremendous diversity in sperm morphology observed among animals likely heralds similar (or potentially greater) diversity in female reproductive organ dimensions (e.g. [10]). By contrast, despite operating in the same environment as fertile sperm, we found that non-fertile sperm evolved slower than female reproductive organ lengths in *Drosophila*. This result is consistent with previous research demonstrating that non-fertile sperm show no evolutionary relationship to female reproductive organ size [25,26]. Thus, our analyses reinforce the notion that the female reproductive tract exerts selection on sperm—but crucially we show that this is only when the sperm perform a fertilization function, even though the non-fertile sperm may also mediate fertility and the outcome of sperm competition (e.g. [23]).

Sperm components can exhibit alternative evolutionary trajectories and patterns of genetic covariance [15,16]. For example, in passerine birds, head length showed different evolutionary trajectories than midpiece or flagellum length [12,18]. For *Drosophila*, we were able to compare evolutionary rates on different subunits of sperm morphology, which provides insight into how selection acts across a functional unit. Contrary to our expectation, fertile sperm head length evolved faster than flagellum length. By contrast, non-fertile sperm showed the opposite pattern, with flagellum length evolving faster than head length. These results suggest that the strength of selection or constraints on responses to selection differed between sperm head compared to flagellum length in fertile and non-fertile sperm. However, at least in *Drosophila*, fertile sperm head length and flagellum length evolved faster than non-fertile sperm components overall. Thus, the rates of evolutionary diversification differ both across and within sperm morphs in *Drosophila*. Previous work on the genetic architecture of fertile and non-fertile sperm in *D. pseudoobscura*, a member of the *obscura* group, found that within each sperm morph, head and flagellum lengths were both positively phenotypically and genetically correlated, but between morphs there was no genetic correlation [16]. Therefore, the alternative rates of evolution observed between sperm morphs are less surprising than the divergence in evolutionary rates observed within morphs. Subunit sperm morphologies were not available for the Lepidoptera evaluated here so a similar analysis could not be performed. However, examining whether differences in evolutionary rates among sperm components exist in a wider range of species, including in more sperm heteromorphic species, would aid in determining whether alternative evolutionary trajectories of sperm components is a general phenomenon.

While we argued above that it is the fertilizing function of sperm that is important in responding to selection, we can surmise additional sources of selection based on variation in evolutionary rates. For example, while the flagellum is thought

**Table 2.** Comparisons of evolutionary rates assuming trait covariance in the observed rate matrices of fertile (eusperm) and non-fertile (parasperm) head length ($L_{head}$) and flagellum length ($L_{flagellum}$) (a–d) and fertile and non-fertile sperm length and seminal receptacle (SR) length (e–h) in *Drosophila* from the *obscura* group. The sample size (*n*), observed ($\sigma^2_{obs}$) and common ($\sigma^2_{common}$) rate matrices are shown for each analysis. The evolutionary rates in the diagonal of the rate matrix is presented in italics as these illustrate difference or similarities in the observed evolutionary rates among traits and how they differ from the common evolutionary rate (presented in brackets). The log-likelihood values for the observed ($\text{Log}L_{obs}$) and common models ($\text{Log}L_{common}$), log-likelihood ratio tests (LRT), with associated degrees of freedom (df), comparing models of observed rates with evolutionarily constrained models where all traits evolve at a common rate, *p*-values and AIC values for the observed ($\text{AIC}_{obs}$) and common ($\text{AIC}_{common}$) models are presented. Significant *p*-values are presented in italic text.

| trait | n | $\sigma^2_{obs}$ ($\sigma^2_{common}$) | $\sigma^2_{obs}$ ($\sigma^2_{common}$) | $\text{Log}L_{obs}$ ($\text{Log}L_{common}$) | $\text{LRT}_{(df)}$ | p | $\text{AIC}_{obs}$ ($\text{AIC}_{common}$) |
|---|---|---|---|---|---|---|---|
| *(a)* | | fertile $L_{head}$ | fertile $L_{flagellum}$ | | | | |
| fertile $L_{head}$ | 12 | 2.15 (1.63) | — | 4.86 (1.94) | 5.84 (df = 1) | 0.02 | −1.72 (2.11) |
| fertile $L_{flagellum}$ | | 1.39 (1.38) | 1.10 (1.63) | | | | |
| *(b)* | | non-fertile $L_{head}$ | non-fertile $L_{flagellum}$ | | | | |
| non-fertile $L_{head}$ | 12 | 0.10 (0.27) | — | 20.00 (16.56) | 6.88 (df = 1) | <0.01 | −31.99 (−27.11) |
| non-fertile $L_{flagellum}$ | | 0.09 (0.09) | 0.43 (0.27) | | | | |
| *(c)* | | fertile $L_{head}$ | non-fertile $L_{head}$ | | | | |
| fertile $L_{head}$ | 12 | 2.15 (1.13) | — | 9.89 (−1.36) | 22.50 (df = 1) | <0.001 | −11.78 (8.72) |
| non-fertile $L_{head}$ | | −0.17 (−0.17) | 0.10 (1.13) | | | | |
| *(d)* | | fertile $L_{flagellum}$ | non-fertile $L_{flagellum}$ | | | | |
| fertile $L_{flagellum}$ | 12 | 1.10 (0.77) | — | 6.76 (4.96) | 3.59 (df = 1) | 0.06 | −5.51 (−3.92) |
| non-fertile $L_{flagellum}$ | | 0.39 (0.39) | 0.43 (0.77) | | | | |
| *(e)* | | fertile $L_{head}$ | seminal receptacle | | | | |
| fertile $L_{head}$ | 11 | 2.07 (1.76) | — | 3.75 (2.59) | 2.32 (df = 1) | 0.13 | 0.51 (0.83) |
| seminal receptacle | | 1.61 (1.61) | 1.45 (1.76) | | | | |
| *(f)* | | fertile $L_{flagellum}$ | seminal receptacle | | | | |
| fertile $L_{flagellum}$ | 11 | 1.18 (1.32) | — | 2.13 (1.94) | 0.38 (df = 1) | 0.53 | 3.74 (2.11) |
| seminal receptacle | | 1.09 (1.09) | 1.45 (1.32) | | | | |
| *(g)* | | non-fertile $L_{head}$ | seminal receptacle | | | | |
| non-fertile $L_{head}$ | 11 | 0.11 (0.78) | — | 9.55 (1.62) | 15.86 (df = 1) | <0.001 | −11.10 (2.76) |
| seminal receptacle | | −0.15 (−0.15) | 1.45 (0.78) | | | | |
| *(h)* | | non-fertile $L_{flagellum}$ | seminal receptacle | | | | |
| non-fertile $L_{flagellum}$ | 11 | 0.44 (0.95) | — | 1.37 (−0.55) | 3.85 (df = 1) | 0.0499 | 5.26 (7.10) |
| seminal receptacle | | 0.13 (0.13) | 1.45 (0.95) | | | | |

to be critical to competitive fertilization success, we find that selection operates much more strongly on fertile sperm heads than flagellum. This pattern could be due to genetic covariance between sperm flagellum and seminal receptacle length (e.g. [41]), leaving less scope for evolutionary diversification in flagellum compared to head length. Alternatively, in *Drosophila* the sperm head interacts with the egg micropyle, and, unlike reproduction in other internal fertilizing species, the sperm head membrane does not lyse with the egg membrane [42]. Therefore, the faster rate of evolution of fertilizing sperm heads compared to flagellum, along with the lack of a relationship between fertile sperm flagellum length and measures of sperm competition strength [25], suggests that selection may act primarily at the interface of the sperm head and entrance to the micropyle during fertilization. To date there is almost nothing known about this interaction. Our work suggests that such a study, taking a phylogenetic approach, would be profitable. By contrast, selection was stronger on flagellum length in non-fertile sperm in *Drosophila*. Non-fertile sperm function in *D. pseudoobscura* to protect brother fertile sperm from spermicide in the female uterus [22], although exactly how non-fertile sperm perform this function remains unknown. Our results hint at the possibility that the non-fertile flagellum may be important in mediating such postcopulatory processes in sperm heteromorphic *Drosophila*, akin to the role of the sperm monomorphic flagellum in influencing sperm competitiveness in sperm monomorphic *Drosophila* species [28].

By focusing on sperm heteromorphic groups, in which one sperm type fertilizes eggs and the other does not, our results provide repeated support for the hypothesis that sperm evolve rapidly in response to their functional role in securing male fertility. Our work also tests if the rapid diversification of sperm is due to the role of sperm length, primarily the flagella, in mediating the outcome of postcopulatory sexual selection. In contrast with this prediction, we identified that fertile sperm heads evolve faster than flagella. Future work should examine the evolutionary diversification of sperm components and their association with the fertilization environment in sperm monomorphic taxa to determine whether such differences are a common response to selection. Finally, the faster rates of diversification detected in fertile sperm match rapid diversification rates in the length of the female reproductive tract. As the female reproductive tract is the putative selective force driving sperm diversification in internal fertilizers generally, our findings suggest that an important next step is assessing how rapidly the female reproductive tract evolves across animals.

Data accessibility. The dataset and code associated with this study are available from the Dryad Digital Repository: https://doi.org/10.5061/dryad.0k6djh9X9 [43].

Authors' contributions. J.L.F. and R.R.S conceived the project; J.L.F, C.D.B. and R.R.S collected data; J.L.F and C.D.B analysed the data; J.L.F, C.D.B. and R.R.S wrote the manuscript.

Competing interests. We declare we have no competing interests.

Funding. This work was supported by a Knut and Alice Wallenberg Academy Fellowship (2016–0146) and a Swedish Research Council grant (2017–04680) to J.L.F, and NSF-DEB (0093149) and a Royal Society Leverhulme Trust Senior Research Fellowship to R.R.S.

Acknowledgements. We thank Matthew Gage and Ted Morrow for generating the Lepidoptera data analysed in this study, and Alessandro Devigili, Ariel Kahrl and three anonymous reviewers for helpful comments on the manuscript.

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
