## [Reviewer comments · Proceedings of the Royal Society B: Biological Sciences]

Review History

RSPB-2019-1477.R0 (Original submission)

Review form: Reviewer 1

Recommendation

Accept with minor revision (please list in comments)

Scientific importance: Is the manuscript an original and important contribution to its field?

Excellent

General interest: Is the paper of sufficient general interest?

Good

Quality of the paper: Is the overall quality of the paper suitable?

Good

Is the length of the paper justified?

Yes

Should the paper be seen by a specialist statistical reviewer?

No

Do you have any concerns about statistical analyses in this paper? If so, please specify them explicitly in your report.

No

It is a condition of publication that authors make their supporting data, code and materials available - either as supplementary material or hosted in an external repository. Please rate, if applicable, the supporting data on the following criteria.

Is it accessible?

Yes

Is it clear?

Yes

Is it adequate?

Yes

Do you have any ethical concerns with this paper?

No

Comments to the Author

Reviewer Comments

In this comparative study the authors provide compelling evidence for the similarity in evolutionary rates of sperm morphology in two sperm heteromorphic insect groups, which are in turn linked to sperm function (fertilizing or not). In addition, the authors compare these evolutionary rates of sperm trait diversification with those of other key reproductive and somatic traits. The paper reads well, the statistical analyses sound, and the study's obvious importance to the field of evolutionary biology is well described and supported. However, at times it lacks clarity/important details for some analytical choices and in the presentation of the, necessarily complex, statistical results. Below, I list the main issues that should be addressed.

Main issues

- It is important to briefly list (e.g. in Intro L119) and later (e.g. in Methods L164) adequately describe and, very importantly, justify the choices of all the non-sperm traits used in the analyses. In particular, the choice of somatic traits. For instance, is Lepidopteran forewing length also restricted to males like the thorax length? Are these somatic traits chosen for their negligible or important role in (pre-copulatory) sexual selection? The reader should not have to read the ESM to get this important methodological information. I assume this was due to word limits. You can perhaps move some more of the technical details (i.e. explaining Ref #46) to Supplementary and return these key biologically relevant aspects of the Methods to the main text.
- I was wondering the need to include the butterflies in this study, since you have no data on female reproductive structure size. Perhaps include only Moths for the main paper and have the Lepidoptera and Butterfly analyses as supplementary material? I assume that there was a reason to compare evolutionary rates of sperm morphology with reproductively meaningful sex-specific traits in both sexes (based on the details in ESM Table S2). If so, the story is rather incomplete for butterflies and Moths make a strong case by themselves for 'convergent evolution' with the *Drosophila* group.
- Tables 1 and 2 are unduly complex. Perhaps for each combination of 2 (or 4) traits, the evolutionary rates should be in the same column instead of a 2x2 (or 4x4) matrix where the top off-diagonal is left blank because it is a repeat of the bottom off-diagonal. It would lengthen the table, yes, but into a more portrait cf. landscape format. I also suggest placing the relevant 'pairwise analyses' to the right of the respective evolutionary rate estimation models outputs. It

will make it easier to follow the results section and the plots.

Other issues

L1: the title should stress that we are working on sperm heteromorphic groups. I'm not sure 'rapid' is warranted

L22: 'two taxa' makes me think of species. Perhaps 'two (insect) groups' is better.

L110: add a line to briefly explain what the different functions of the two sperm types are (i.e. what the non-fertile sperm is for)

L119: you say here the somatic traits chosen are a proxy for naturally selected traits sensu Ref#5. Are the sizes of female reproductive tract component also such traits? Or were you referring to (male?) wing and thorax. Be more explicit.

L164: here you list 'sperm', 'reproductive' and 'somatic' type of trait for which you will calculate evolutionary rates. Now we have three types. But as mentioned earlier, there is no detailed description and justification for all the ones used in the analyses

L202: what are these taxonomic groups? Spell them out.

L207: I think that once the evolutionary rates are in the same column vs. in a matrix format in the Table, it will make it easier to see how one is 'slower' or 'faster' and by how much.

L233&L298: don't forget to add the measure of the trait e.g. 'receptacle length'. It is important because mass or volume would make the reader think of sperm numbers being more relevant than sperm length, for instance.

L234-L239: Why are these results in the text cf. Table?

L292-293: How does male body size (or the other 'somatic' traits in the study) 'play a direct role in securing male fertility'. The latter should have been made clearer at the point that the choice of traits is justified.

Table 1 (&2): Are the pairwise comparisons using fertile & non-fertile sperm referent to total sperm length in all cases and for all taxonomic groups? Perhaps be clearer in the Table 2: What is the superscript 'a' referent to in Moths and Butterflies?

Review form: Reviewer 2

Recommendation

Major revision is needed (please make suggestions in comments)

Scientific importance: Is the manuscript an original and important contribution to its field?

Excellent

General interest: Is the paper of sufficient general interest?

Excellent

Quality of the paper: Is the overall quality of the paper suitable?

Good

Is the length of the paper justified?

Yes

Should the paper be seen by a specialist statistical reviewer?

Yes

Do you have any concerns about statistical analyses in this paper? If so, please specify them explicitly in your report.

No

It is a condition of publication that authors make their supporting data, code and materials available - either as supplementary material or hosted in an external repository. Please rate, if applicable, the supporting data on the following criteria.

Is it accessible?

Yes

Is it clear?

Yes

Is it adequate?

Yes

Do you have any ethical concerns with this paper?

No

Comments to the Author

Major comments:

This paper examines the hypothesis that fertile sperm traits evolve more quickly than other male traits (because of their function in fertilization and therefore effect on male reproductive success). To examine this hypothesis in a powerful way, the authors compile comparative data on sperm (total or head and flagellum) length, female storage organ length, and body (either thorax or forewing) length in two groups of insects (*Drosophila* spp. from the *obscura* group and *Lepidoptera*). Their analyses then compare the rates of trait evolution for fertile sperm length, non-fertile sperm length, female reproductive tract length and somatic length to ask whether fertile sperm length evolves faster than other reproductive and somatic lengths. The authors argue that by comparing how these two sperm types evolve – since they are generated in the same tissues and experience the same environment during selection- allows a powerful examination of this hypothesis that sperm traits evolve faster than other male traits.

I think the analyses presented in this paper are powerful and interesting. I love the idea of comparing the evolution of sperm traits for those sperm involved directly in fertilization (i.e. eusperm or eupyrene sperm) and those not involved directly in fertilization (i.e. parasperm or apyrene sperm). The comparison with the rate of female reproductive tract evolution as well as somatic length trait is also interesting, timely and informative. I am therefore very much in support of publishing this paper and the analyses it reports.

There are, however, a few things about the framing of the paper that I feel need improvement for greater clarity and precision.

1) The comparison of the rate of evolution of the two sperm types really focuses on comparing sperm that are involved in fertilization and those that are not. But the framing language is about sperm traits and other male traits. Since both are sperm, but only one type functions in the typical way we think about sperm, this gets a bit confusing. The paper should be clear that the comparison is really about traits involved directly in fertilization or not. And acknowledge that the non-fertile sperm are still likely involved indirectly in fertilization and therefore also experiencing post-copulatory sexual selection. The authors could also make the impact of their work greater, if they help the reader understand why comparing fertile and non-fertile sperm types is different than comparing fertile sperm with other traits directly related to differential fertilization success and the outcome of sperm competition (e.g. testes size, other ejaculate characteristics, etc.).

2) In my opinion, the real power of the analyses is in looking at the correlated evolution of the two sperm types, along with the female reproductive tract and a “comparable” somatic length variable. Yet, the female reproductive tract analyses seem like a bit of a “throwaway” in the

abstract and introduction as written.

3) There are a variety of reasons that two traits might evolve at different rates. This is not discussed in the introduction or discussion. And the discussion makes inferences about the strength/form of selection on the traits based on differences in the rate of trait evolution without acknowledging the assumptions being made. The rate of trait evolution could also easily differ if these traits have different heritability, genetic correlations, evolutionary constraints, or trade-offs. These other reasons that the rates could differ must be discussed explicitly and the assumptions being made must be acknowledged and fully transparent in the introduction and discussion.

Minor comments:

The “readability” of the paper would benefit from you using the terms fertilizing and non-fertilizing sperm or fertile and non-fertile sperm throughout (the more precise terms of eusperm/parasperm or eupyrene/apyrene sperm could also be included in parentheses where useful). This would especially be nice to have in the Figures and Tables to remind the reader which is which.

Abstract:

Line 29-30: “exhibit contrasting rates of divergence.” This phrasing felt awkward to me. Perhaps state more directly “differ in their rates of divergence”

Line 32-33: An admittedly picky point, but your data can’t say anything directly about whether sperm traits evolved “because of their functional role in securing fertility”. You could say we provide direct evidence that sperm length evolves more rapidly in fertile sperm, likely because of....

Line 34-35: An interesting result and yet this is the first time you mention female trait evolution and so it feels like it comes out of nowhere.

Introduction:

Lines 40-45: I believe I understand what you mean here but had to think about it a bit. And the challenge stems from many things- not just the one you mention. I suggest you rephrase this.

Lines 52-55: This is a really key point for your paper. I suggest expanding this explanation a bit more for the reader to explain what is confounding the interpretation of these rates of evolution. I am not at all disagreeing with your statement- just suggesting that it is a key argument that warrants further explanation since the importance of this contribution depends on the reader understanding this point.

Line 80-81 Why? I do not disagree- instead I suggest you make the argument explicitly, rather than just state that it is so. Why are sperm traits expected to evolve faster than other sexually selected traits? Is it because sexual selection is stronger on them (because they have a larger effect on fitness), because they are somehow more evolvable (less constrained genetically or otherwise), have few or weaker natural selection/sexual selection tradeoffs, or have higher heritability/underlying genetic variation, or because of the combination of these things?

You may want to be explicit that you use natural selection to capture all of other forms of selection- e.g. viability and fecundity-since there is some disagreement about whether sexual selection is a type of natural selection or a distinct form of selection- which is irrelevant to your claim but could confuse or annoy some readers.

Line 111: See my general comment above. I think we need a bit more clarification why this is better than past analyses on multiple sexually selected male traits.

Lines 117-120: I think this is the key strength of this work and yet this is the first the reader hears about it and why female reproductive trait evolution is worth including.

Methods:

I suggest including more concrete information about the actual variables included in the analyses within the main methods section of the paper. I felt there was insufficient information on how the other reproductive and somatic traits were chosen and how the data on them were assembled. This all made much more sense once I read the supplemental materials (and could see that only length was compared for various traits).

Given that both groups are monophyletic, please explain why you split the Lepidoptera into three groups? Were the results qualitatively the same if you treated Lepidoptera as one group?

This section would benefit from having stated objectives and explanation of how the data and analyses relate to those objectives.

Lines 135-142 (esp. 141-142)- this is a very clear explanation of the comparison and its power. I suggest finding a place to make the comparison this clear (though in more general form) in the introduction- especially that the comparison is really otherwise same or extremely similar traits and environments but only one of which serves to fertilize the egg.

Line 164-165 (and in general) I found the reference to “sperm morphology”, “reproductive traits” “somatic traits” confusing and misleading given that only a few specific length traits are in the dataset. Why not refer specifically to sperm length, female sperm storage organ length and body length traits- or something like that- to be clear what traits are being compared.

The comparative methods used in this paper are not my area of expertise, especially the methods for comparing evolutionary rates. I found the description of the methods used clear and relatively straight-forward, though readers may not all find the sigma square parameter intuitive.

Results

Line 211: It would be more correct to say “sperm length evolves faster” not “sperm evolve faster” (here and throughout the paper). This is also true for length of the sperm storage organ, etc.

Line 221: In the next few lines, the observed patterns are clear, but I wondered what “opposing evolutionary rates” meant- and my initial assumption was that you meant differences in the direction of trait evolution rather than a difference in the rate of evolution.

Line 227- That is a truly dramatic difference in trait evolution between fertile and non-fertile sperm length.

Line 231: Like with “opposing” I found “evolve at distinct rates” a bit confusing. Do you mean the magnitude of the evolutionary rate parameters were statistically different for the same traits? If so, it seems more correct to say the evolutionary rates differed significantly.

Discussion:

Line 273 and throughout the paper- What does it mean to say the patterns are convergent? I am more familiar with this term meaning the independent origin of the same traits in different groups rather than similar rates of evolution. I therefore don't think convergent is the best word

to capture your results.

Line 277- I suggest writing “different rates of trait evolution instead of “rates of phenotypic diversity across traits” (or otherwise rewriting this statement) – since I don’t really understand what that phrase is saying as written.

Line 281- I do not agree that you can say this about “alternative selection pressures” based on your data – at least not so strongly. They evolve at different rates and there are a variety of other reasons that might happen (e.g. different heritabilities, genetic correlations, trade-offs, different strength of same selection pressure, evolutionary constraints, etc.).

Line 289- Lines 282-290 are a powerful explanation of the results and their relevance, but- and this is an admittedly picky point- I find “dramatically different rates of phenotypic diversification” overstated given that some differences are large but others are small.

Lines 316-319- These patterns are very interesting and in general that the evolutionary pattern of sperm and female tracts is one of the key strengths of this set of analyses. I suggest introducing this /setting this up a more in the introduction.

Line 328 and throughout the discussion- there are various key assumptions being made (e.g. same heritability/genetic covariances among traits) that needs to be acknowledged when you make any inference about what the rate of trait evolution says about selection pressures. I find this interpretation over-stated given that there are many other possible reasons besides the strength of selection for differences in trait evolution.

Line 373- As reviewed in the introduction, a general explanation exists but has been hard to test.

Tables 1 and 2- Great information. But it is hard to see what the significant statistical differences “mean” about the relative absolute rate and direction of evolution. The summary at the bottom is very useful. I wonder, however, if the direction of difference could be given. For example, replace the versus with greater than, less than or approx. equal symbols (or words if preferred).

Review form: Reviewer 3 (Patricia Brennan)

Recommendation

Accept with minor revision (please list in comments)

Scientific importance: Is the manuscript an original and important contribution to its field?

Excellent

General interest: Is the paper of sufficient general interest?

Excellent

Quality of the paper: Is the overall quality of the paper suitable?

Good

Is the length of the paper justified?

Yes

Should the paper be seen by a specialist statistical reviewer?

Yes

Do you have any concerns about statistical analyses in this paper? If so, please specify them explicitly in your report.

No

It is a condition of publication that authors make their supporting data, code and materials available - either as supplementary material or hosted in an external repository. Please rate, if applicable, the supporting data on the following criteria.

Is it accessible?

Yes

Is it clear?

Yes

Is it adequate?

Yes

Do you have any ethical concerns with this paper?

No

Comments to the Author

The authors report on their test of the idea that sperm evolve faster than other male traits, by comparing the rates of phenotypic divergence of non fertile and fertile sperm in two taxa. Their argument is that fertile and non fertile sperm are produced by the same tissue, and therefore are in fact more comparable, than examining traits that are not produced by the same tissue, or encounter the same selective environment. They also compare rates of evolution of sperm components, and female reproductive tracts.

In line with their predictions, they find that fertile sperm evolve faster than non fertile sperm, and faster than body size, but at the same rate than female reproductive tract.

They also find that the head evolves faster in the fertile sperm, while the flagellum evolves faster in non fertile sperm.

I like the motivation of the paper, and there are interesting repercussions for proposing this novel way of examining sperm evolution. The paper is generally well written (some minor suggestions follow), but clarification is needed in places.

I am not familiar with the analysis, so I cannot comment as to whether they have been deployed properly, so having a reviewer who knows these methods would be important.

Line 47: change "is" for "are"

Line 48: insert comma after sexual selection

Lines 52-55: can you clarify the different selective episodes you are referring to here?

Line 72: after per se, please elaborate why. Perhaps providing an example would be useful.

Line 83-85: I don't think this is true anymore. Perhaps simply starting this paragraph with the next sentence would avoid the need for this generalization.

Lines 145-147: A recent paper by Sakai et al 2019 in PNAS showed that parasperm in *B. mori* (lepidoptera) is necessary for sperm migration to the female organs, offering strong evidence that parasperm may not be cheap fillers. These data I think may change how you interpret some of your results in the discussion.

Line 167: Insert sample sizes here please. These are unclear in the results. How many individuals were measured to obtain estimates per species? are these averages? Also insert reduced sample sizes at the end of line 169.

Line 176-181: I suggest deleting this last few sentences since the next section explains this again. Add "Supplementary methods" to the parenthesis that says ESM.

Line 228-229: replace “was on the border of statistical significance” with the actual value.

Line 233: What measurements of the seminal receptacle did you analyze? Length? volume? In the figure it says length so please clarify if that is it. The spermathecal duct and the seminal receptacle have different functions in these two taxa. The seminal receptacle stores sperm, whereas the duct is the passageway to the spermatheca, so how does the different function of these organs affect your conclusions?

Line 246: Why does forewing length evolve faster than thorax length? Do you have data for thorax length in lepidoptera that would be more comparable? forewing length may be under sexual selection.

Line 268: Here it says the rate of evolution of forewing length is the same as infertile sperm, but in line 246, it says that forewing length evolve at an intermediate rate between fertile and infertile sperm. Please clarify that you are comparing moths vs. butterflies in the text.

Line 301-319: This needs clarification. “Female reproductive tract” is not interchangeable with spermathecal duct length or seminal receptacle length. The female reproductive tract has many components that have different functions in addition to interacting with male reproductive components, and even those interactions can be quite diverse: Interacting with the spermatophore, the aedagus, or the ejaculate, so clarifying the role you are referring to when it comes to sperm interactions would seem important here. This section needs rewriting. At the end, again I think it is important to consider the new evidence provided by Sakait et al 2019 I reconsidering what the role of infertile sperm may be in lepidoptera.

Line 352-354. This requires more explanation. Are you suggesting that sperm fit just right in the female tract so if they were longer they would not fit?

Line 357-361: Cool idea!

You might want to cite Higginson and Pitnick (2011) for a summary of the possible functions of parasperm.

Decision letter (RSPB-2019-1477.R0)

01-Aug-2019

Dear Dr Fitzpatrick:

I am writing to inform you that your manuscript RSPB-2019-1477 entitled "Convergent evidence for the rapid evolution of sperm morphology" has, in its current form, been rejected for publication in Proceedings B.

This action has been taken on the advice of referees, who have recommended that substantial revisions are necessary. With this in mind we would be happy to consider a resubmission, provided the comments of the referees are fully addressed. However please note that this is not a provisional acceptance.

Sincerely,
 Professor Gary Carvalho
 mailto: proceedingsb@royalsociety.org

Associate Editor
 Board Member: 1
 Comments to Author:
 Dear Authors,

I have now received three careful and constructive reviews for your manuscript "Convergent evidence for the rapid evolution of sperm morphology". All reviewers see merit in the topic and feel that the manuscript can be of broad interest to the general audience at Proc B. However they each also raise many issues that need to be addressed before that decision can be made. For example, although the concept and experimental design of the experiment are strong, the authors should simplify the result tables, better justify the analyses used and parameters chosen (as they pertain to the main objectives: see comments by Reviewer 1) and provide better descriptions and explanations of the results (i.e. impact, why are the main findings important?). They should also revise the framing of the introduction (see major comments by Reviewer 2), and be careful and consistent with their sperm-related terminology. I hope that you find the comments of the reviewers helpful in revising your manuscript.

Reviewer(s)' Comments to Author:

Referee: 1

Comments to the Author(s)
 Reviewer Comments

In this comparative study the authors provide compelling evidence for the similarity in evolutionary rates of sperm morphology in two sperm heteromorphic insect groups, which are in turn linked to sperm function (fertilizing or not). In addition, the authors compare these evolutionary rates of sperm trait diversification with those of other key reproductive and somatic traits. The paper reads well, the statistical analyses sound, and the study's obvious importance to the field of evolutionary biology is well described and supported. However, at times it lacks clarity/important details for some analytical choices and in the presentation of the, necessarily complex, statistical results. Below, I list the main issues that should be addressed.

Main issues

- It is important to briefly list (e.g. in Intro L119) and later (e.g. in Methods L164) adequately describe and, very importantly, justify the choices of all the non-sperm traits used in the analyses. In particular, the choice of somatic traits. For instance, is Lepidopteran forewing length also restricted to males like the thorax length? Are these somatic traits chosen for their negligible or important role in (pre-copulatory) sexual selection? The reader should not have to read the ESM to get this important methodological information. I assume this was due to word limits. You can perhaps move some more of the technical details (i.e. explaining Ref #46) to Supplementary and return these key biologically relevant aspects of the Methods to the main text.
- I was wondering the need to include the butterflies in this study, since you have no data on female reproductive structure size. Perhaps include only Moths for the main paper and have the Lepidoptera and Butterfly analyses as supplementary material? I assume that there was a reason to compare evolutionary rates of sperm morphology with reproductively meaningful sex-specific traits in both sexes (based on the details in ESM Table S2). If so, the story is rather incomplete for butterflies and Moths make a strong case by themselves for 'convergent evolution' with the *Drosophila* group.
- Tables 1 and 2 are unduly complex. Perhaps for each combination of 2 (or 4) traits, the evolutionary rates should be in the same column instead of a 2x2 (or 4x4) matrix where the top off-diagonal is left blank because it is a repeat of the bottom off-diagonal. It would lengthen the table, yes, but into a more portrait cf. landscape format. I also suggest placing the relevant 'pairwise analyses' to the right of the respective evolutionary rate estimation models outputs. It will make it easier to follow the results section and the plots.

Other issues

- L1: the title should stress that we are working on sperm heteromorphic groups. I'm not sure 'rapid' is warranted
- L22: 'two taxa' makes me think of species. Perhaps 'two (insect) groups' is better.
- L110: add a line to briefly explain what the different functions of the two sperm types are (i.e. what the non-fertile sperm is for)
- L119: you say here the somatic traits chosen are a proxy for naturally selected traits sensu Ref#5. Are the sizes of female reproductive tract component also such traits? Or were you referring to (male?) wing and thorax. Be more explicit.
- L164: here you list 'sperm', 'reproductive' and 'somatic' type of trait for which you will calculate evolutionary rates. Now we have three types. But as mentioned earlier, there is no detailed description and justification for all the ones used in the analyses
- L202: what are these taxonomic groups? Spell them out.
- L207: I think that once the evolutionary rates are in the same column vs. in a matrix format in the Table, it will make it easier to see how one is 'slower' or 'faster' and by how much.
- L233&L298: don't forget to add the measure of the trait e.g. 'receptacle length'. It is important because mass or volume would make the reader think of sperm numbers being more relevant than sperm length, for instance.
- L234-L239: Why are these results in the text cf. Table?
- L292-293: How does male body size (or the other 'somatic' traits in the study) 'play a direct role in securing male fertility'. The latter should have been made clearer at the point that the choice of traits is justified.
- Table 1 (&2): Are the pairwise comparisons using fertile & non-fertile sperm referent to total sperm length in all cases and for all taxonomic groups? Perhaps be clearer in the Table 2: What is the superscript 'a' referent to in Moths and Butterflies?

Referee: 2

Comments to the Author(s)

Major comments:

This paper examines the hypothesis that fertile sperm traits evolve more quickly than other male traits (because of their function in fertilization and therefore effect on male reproductive success). To examine this hypothesis in a powerful way, the authors compile comparative data on sperm (total or head and flagellum) length, female storage organ length, and body (either thorax or forewing) length in two groups of insects (*Drosophila* spp. from the *obscura* group and *Lepidoptera*). Their analyses then compare the rates of trait evolution for fertile sperm length, non-fertile sperm length, female reproductive tract length and somatic length to ask whether fertile sperm length evolves faster than other reproductive and somatic lengths. The authors argue that by comparing how these two sperm types evolve – since they are generated in the same tissues and experience the same environment during selection- allows a powerful examination of this hypothesis that sperm traits evolve faster than other male traits.

I think the analyses presented in this paper are powerful and interesting. I love the idea of comparing the evolution of sperm traits for those sperm involved directly in fertilization (i.e. eusperm or eupyrene sperm) and those not involved directly in fertilization (i.e. parasperm or apyrene sperm). The comparison with the rate of female reproductive tract evolution as well as somatic length trait is also interesting, timely and informative. I am therefore very much in support of publishing this paper and the analyses it reports.

There are, however, a few things about the framing of the paper that I feel need improvement for greater clarity and precision.

- 1) The comparison of the rate of evolution of the two sperm types really focuses on comparing sperm that are involved in fertilization and those that are not. But the framing language is about sperm traits and other male traits. Since both are sperm, but only one type functions in the typical way we think about sperm, this gets a bit confusing. The paper should be clear that the comparison is really about traits involved directly in fertilization or not. And acknowledge that the non-fertile sperm are still likely involved indirectly in fertilization and therefore also experiencing post-copulatory sexual selection. The authors could also make the impact of their work greater, if they help the reader understand why comparing fertile and non-fertile sperm types is different than comparing fertile sperm with other traits directly related to differential fertilization success and the outcome of sperm competition (e.g. testes size, other ejaculate characteristics, etc.).
- 2) In my opinion, the real power of the analyses is in looking at the correlated evolution of the two sperm types, along with the female reproductive tract and a “comparable” somatic length variable. Yet, the female reproductive tract analyses seem like a bit of a “throwaway” in the abstract and introduction as written.
- 3) There are a variety of reasons that two traits might evolve at different rates. This is not discussed in the introduction or discussion. And the discussion makes inferences about the strength/form of selection on the traits based on differences in the rate of trait evolution without acknowledging the assumptions being made. The rate of trait evolution could also easily differ if these traits have different heritability, genetic correlations, evolutionary constraints, or trade-offs. These other reasons that the rates could differ must be discussed explicitly and the assumptions being made must be acknowledged and fully transparent in the introduction and discussion.

Minor comments:

The “readability” of the paper would benefit from you using the terms fertilizing and non-fertilizing sperm or fertile and non-fertile sperm throughout (the more precise terms of eusperm/parasperm or eupyrene/apyrene sperm could also be included in parentheses where useful). This would especially be nice to have in the Figures and Tables to remind the reader which is which.

Abstract:

Line 29-30: “exhibit contrasting rates of divergence.” This phrasing felt awkward to me. Perhaps state more directly “differ in their rates of divergence”

Line 32-33: An admittedly picky point, but your data can’t say anything directly about whether sperm traits evolved “because of their functional role in securing fertility”. You could say we provide direct evidence that sperm length evolves more rapidly in fertile sperm, likely because of....

Line 34-35: An interesting result and yet this is the first time you mention female trait evolution and so it feels like it comes out of nowhere.

Introduction:

Lines 40-45: I believe I understand what you mean here but had to think about it a bit. And the challenge stems from many things- not just the one you mention. I suggest you rephrase this.

Lines 52-55: This is a really key point for your paper. I suggest expanding this explanation a bit more for the reader to explain what is confounding the interpretation of these rates of evolution. I am not at all disagreeing with your statement- just suggesting that it is a key argument that warrants further explanation since the importance of this contribution depends on the reader understanding this point.

Line 80-81 Why? I do not disagree- instead I suggest you make the argument explicitly, rather than just state that it is so. Why are sperm traits expected to evolve faster than other sexually selected traits? Is it because sexual selection is stronger on them (because they have a larger effect on fitness), because they are somehow more evolvable (less constrained genetically or otherwise), have few or weaker natural selection/sexual selection tradeoffs, or have higher heritability/underlying genetic variation, or because of the combination of these things?

You may want to be explicit that you use natural selection to capture all of other forms of selection- e.g. viability and fecundity-since there is some disagreement about whether sexual selection is a type of natural selection or a distinct form of selection- which is irrelevant to your claim but could confuse or annoy some readers.

Line 111: See my general comment above. I think we need a bit more clarification why this is better than past analyses on multiple sexually selected male traits.

Lines 117-120: I think this is the key strength of this work and yet this is the first the reader hears about it and why female reproductive trait evolution is worth including.

Methods:

I suggest including more concrete information about the actual variables included in the analyses within the main methods section of the paper. I felt there was insufficient information on how the other reproductive and somatic traits were chosen and how the data on them were assembled. This all made much more sense once I read the supplemental materials (and could see that only length was compared for various traits).

Given that both groups are monophyletic, please explain why you split the Lepidoptera into three groups? Were the results qualitatively the same if you treated Lepidoptera as one group?

This section would benefit from having stated objectives and explanation of how the data and analyses relate to those objectives.

Lines 135-142 (esp. 141-142)- this is a very clear explanation of the comparison and its power. I suggest finding a place to make the comparison this clear (though in more general form) in the introduction- especially that the comparison is really otherwise same or extremely similar traits and environments but only one of which serves to fertilize the egg.

Line 164-165 (and in general) I found the reference to “sperm morphology”, “reproductive traits” “somatic traits” confusing and misleading given that only a few specific length traits are in the dataset. Why not refer specifically to sperm length, female sperm storage organ length and body length traits- or something like that- to be clear what traits are being compared.

The comparative methods used in this paper are not my area of expertise, especially the methods for comparing evolutionary rates. I found the description of the methods used clear and relatively straight-forward, though readers may not all find the sigma square parameter intuitive.

Results

Line 211: It would be more correct to say “sperm length evolves faster” not “sperm evolve faster” (here and throughout the paper). This is also true for length of the sperm storage organ, etc.

Line 221: In the next few lines, the observed patterns are clear, but I wondered what “opposing evolutionary rates” meant- and my initial assumption was that you meant differences in the direction of trait evolution rather than a difference in the rate of evolution.

Line 227- That is a truly dramatic difference in trait evolution between fertile and non-fertile sperm length.

Line 231: Like with “opposing” I found “evolve at distinct rates” a bit confusing. Do you mean the magnitude of the evolutionary rate parameters were statistically different for the same traits? If so, it seems more correct to say the evolutionary rates differed significantly.

Discussion:

Line 273 and throughout the paper- What does it mean to say the patterns are convergent? I am more familiar with this term meaning the independent origin of the same traits in different groups rather than similar rates of evolution. I therefore don't think convergent is the best word to capture your results.

Line 277- I suggest writing “different rates of trait evolution instead of “rates of phenotypic diversity across traits” (or otherwise rewriting this statement) – since I don't really understand what that phrase is saying as written.

Line 281- I do not agree that you can say this about “alternative selection pressures” based on your data – at least not so strongly. They evolve at different rates and there are a variety of other reasons that might happen (e.g. different heritabilities, genetic correlations, trade-offs, different strength of same selection pressure, evolutionary constraints, etc.).

Line 289- Lines 282-290 are a powerful explanation of the results and their relevance, but- and this is an admittedly picky point- I find “dramatically different rates of phenotypic diversification” overstated given that some differences are large but others are small.

Lines 316-319- These patterns are very interesting and in general that the evolutionary pattern of sperm and female tracts is one of the key strengths of this set of analyses. I suggest introducing this /setting this up a more in the introduction.

Line 328 and throughout the discussion- there are various key assumptions being made (e.g.

same heritability/genetic covariances among traits) that needs to be acknowledged when you make any inference about what the rate of trait evolution says about selection pressures. I find this interpretation over-stated given that there are many other possible reasons besides the strength of selection for differences in trait evolution.

Line 373- As reviewed in the introduction, a general explanation exists but has been hard to test.

Tables 1 and 2- Great information. But it is hard to see what the significant statistical differences “mean” about the relative absolute rate and direction of evolution. The summary at the bottom is very useful. I wonder, however, if the direction of difference could be given. For example, replace the versus with greater than, less than or approx. equal symbols (or words if preferred).

Referee: 3

Comments to the Author(s)

The authors report on their test of the idea that sperm evolve faster than other male traits, by comparing the rates of phenotypic divergence of non fertile and fertile sperm in two taxa. Their argument is that fertile and non fertile sperm are produced by the same tissue, and therefore are in fact more comparable, than examining traits that are not produced by the same tissue, or encounter the same selective environment. They also compare rates of evolution of sperm components, and female reproductive tracts.

In line with their predictions, they find that fertile sperm evolve faster than non fertile sperm, and faster than body size, but at the same rate than female reproductive tract.

They also find that the head evolves faster in the fertile sperm, while the flagellum evolves faster in non fertile sperm.

I like the motivation of the paper, and there are interesting repercussions for proposing this novel way of examining sperm evolution. The paper is generally well written (some minor suggestions follow), but clarification is needed in places.

I am not familiar with the analysis, so I cannot comment as to whether they have been deployed properly, so having a reviewer who knows these methods would be important.

Line 47: change “is” for “are”

Line 48: insert comma after sexual selection

Lines 52-55: can you clarify the different selective episodes you are referring to here?

Line 72: after per se, please elaborate why. Perhaps providing an example would be useful.

Line 83-85: I don’t think this is true anymore. Perhaps simply starting this paragraph with the next sentence would avoid the need for this generalization.

Lines 145-147: A recent paper by Sakai et al 2019 in PNAS showed that parasperm in *B. mori* (lepidoptera) is necessary for sperm migration to the female organs, offering strong evidence that parasperm may not be cheap fillers. These data I think may change how you interpret some of your results in the discussion.

Line 167: Insert sample sizes here please. These are unclear in the results. How many individuals were measured to obtain estimates per species? are these averages? Also insert reduced sample sizes at the end of line 169.

Line 176-181: I suggest deleting this last few sentences since the next section explains this again. Add “Supplementary methods” to the parenthesis that says ESM.

Line 228-229: replace “was on the border of statistical significance” with the actual value.

Line 233: What measurements of the seminal receptacle did you analyze? Length? volume? In the figure it says length so please clarify if that is it. The spermathecal duct and the seminal receptacle have different functions in these two taxa. The seminal receptacle stores sperm,

whereas the duct is the passageway to the spermatheca, so how does the different function of these organs affect your conclusions?

Line 246: Why does forewing length evolve faster than thorax length? Do you have data for thorax length in lepidoptera that would be more comparable? forewing length may be under sexual selection.

Line 268: Here it says the rate of evolution of forewing length is the same as infertile sperm, but in line 246, it says that forewing length evolve at an intermediate rate between fertile and infertile sperm. Please clarify that you are comparing moths vs. butterflies in the text.

Line 301-319: This needs clarification. "Female reproductive tract" is not interchangeable with spermathecal duct length or seminal receptacle length. The female reproductive tract has many components that have different functions in addition to interacting with male reproductive components, and even those interactions can be quite diverse: Interacting with the spermatophore, the aedagus, or the ejaculate, so clarifying the role you are referring to when it comes to sperm interactions would seem important here. This section needs rewriting. At the end, again I think it is important to consider the new evidence provided by Sakait et al 2019 I reconsidering what the role of infertile sperm may be in lepidoptera.

Line 352-354. This requires more explanation. Are you suggesting that sperm fit just right in the female tract so if they were longer they would not fit?

Line 357-361: Cool idea!

You might want to cite Higginson and Pitnick (2011) for a summary of the possible functions of parasperm.

Author's Response to Decision Letter for (RSPB-2019-1477.R0)

See Appendix A.

RSPB-2019-2260.R0

Review form: Reviewer 2

Recommendation

Accept as is

Scientific importance: Is the manuscript an original and important contribution to its field?

Excellent

General interest: Is the paper of sufficient general interest?

Good

Quality of the paper: Is the overall quality of the paper suitable?

Excellent

Is the length of the paper justified?

Yes

Should the paper be seen by a specialist statistical reviewer?

No

Do you have any concerns about statistical analyses in this paper? If so, please specify them explicitly in your report.

No

It is a condition of publication that authors make their supporting data, code and materials available - either as supplementary material or hosted in an external repository. Please rate, if applicable, the supporting data on the following criteria.

Is it accessible?

Yes

Is it clear?

Yes

Is it adequate?

Yes

Do you have any ethical concerns with this paper?

No

Comments to the Author

The authors have addressed all of the comments and concerns I raised in my review. My only suggestion, which they should feel free to ignore, is that they reconsider their new title and perhaps instead use their old title but with accelerated in place of rapid.

Review form: Reviewer 4

Recommendation

Major revision is needed (please make suggestions in comments)

Scientific importance: Is the manuscript an original and important contribution to its field?

Good

General interest: Is the paper of sufficient general interest?

Good

Quality of the paper: Is the overall quality of the paper suitable?

Acceptable

Is the length of the paper justified?

Yes

Should the paper be seen by a specialist statistical reviewer?

No

Do you have any concerns about statistical analyses in this paper? If so, please specify them explicitly in your report.

Yes

It is a condition of publication that authors make their supporting data, code and materials available - either as supplementary material or hosted in an external repository. Please rate, if applicable, the supporting data on the following criteria.

Is it accessible?

Yes

Is it clear?

Yes

Is it adequate?

No

Do you have any ethical concerns with this paper?

No

Comments to the Author

This manuscript presents an analysis of sperm cell evolution and provides a test of the principal hypothesis for their extreme morphology diversity. The authors ask three questions:

- 1) Does sperm cell morphology evolve quicker than that of other male traits?
- 2) If so, does the higher rate of evolution in sperm relate to their function in fertilization?
- 3) Finally, do the various components of a sperm cell evolve in concert, or do some components evolve more quickly than others?

The authors recognize a naturally-occurring test of their first two questions in species groups with heteromorphic sperm (i.e. species that produce fertile and non-fertile sperm morphs: moths, butterflies, and *Drosophila obscura*). This is a clever and original insight. By comparing morphological evolution in fertile and non-fertile sperm, which originate in the same tissue and occur in the same part of the body, the authors remove many confounding variables and isolate function as the primary predictor of evolutionary rate. The authors report faster rates of evolution in fertile sperm versus non-fertile sperm and wing length, and the find that rates of evolution in fertile sperm match those of the female reproductive tract, consistent with a role of fertilization in driving sperm evolution. The approach is innovative and the study has the potential to be impactful, but two serious issues should be addressed.

Major Comments

1. Unreliable phylogenies for Lepidopterans. The authors made a concerted effort to use the same or similar genetic sequences to estimate phylogenies in the three species groups (i.e. COII was used to construct and date trees for *Drosophila*, the related COI was used to construct and date trees for butterflies and, separately, moths), but this is a bit misguided, as the greater priority is to use sequences that allow reliable estimates of phylogenies over the timescales in question.

The three species groups used in this study represent different taxonomic ranks and have diversified over vastly different timescales – *Drosophila obscura* is a subgroup within a genus that arose ~15-20 my (Gao et al. 2007, Obbard et al. 2012), while Lepidoptera is an entire order of insects that arose ~190 my (Mitter et al. 2017) and butterflies are a superfamily within Lepidoptera that evolved ~80-140 my (Mitter et al. 2017). Mitochondrial DNA has a relatively high substitution rate, making it useful for estimating relationships among closely-related species. For the *Drosophila obscura* group, COII is likely a good choice, but COI is unreliable for estimating phylogenies among the Lepidopterans, which contain species from different families and superfamilies, due to saturation. The authors note that COI is the only gene available for the moths in their dataset. One potential option is to use published phylogenies to find the topology of species in the dataset and then fix this topology when building a tree in BEAST, thus using

COI only to estimate divergence times. But even then, dates from the more deeply-diverged splits are unlikely to be accurate. A potential fix could be use 2nd and 3rd codon positions, which have lower substitution rates (see references within Obbard et al. 2012). An alternative route may be to drop the moths from the analyses.

It could be argued that this problem is unlikely to affect the key results. The phylogeny will be inaccurate, and the estimated rates of evolution will be inaccurate. But estimated rates will be inaccurate in the same way for all traits. It might not be wrong to conclude that one trait evolves faster than another.

2. A second significant issue is the splitting of Lepidopterans in the dataset into two insect groups: butterflies (a Lepidopteran clade) and moths (non-butterfly Lepidopterans). This appears to have been done because data on the female reproductive tract were only available for moths, but it creates the misleading impression that the evolutionary dynamics uncovered in this analysis (i.e. faster rates of evolution in fertile versus non-fertile sperm) are independently replicated three times in insects. Butterflies are nested within moths and did not evolve heteromorphic sperm independently, and their differential evolutionary rates should not be considered independent either. The valid approach would be to join moths and butterflies into a Lepidopteran tree for the majority of analyses, then prune out the butterflies for the test that includes the female reproductive tract. This change should be reflected in the reporting of the results.

Minor Comments

Abstract:

1. Here and throughout the paper, the term “divergence” (e.g. phenotypic divergence, evolutionary divergence, morphological divergence) is used with reference to whole species groups, and to me this creates some confusion. When I think of divergence I think of the evolution of differences between two lineages, such as sister species that have recently diverged. I recommend replacing divergence with evolution to make it clearer that we’re talking about whole species groups.

2. Line 28-30. This sentence makes it sound as though the only component of sperm morphology that was studied was head length. This more interesting result that should be summarized here is that the two types of sperm have evolved quite differently (faster head evolution in fertile sperm, faster flagellum evolution in non-fertile sperm).

3. Line 33. I think this would be stronger if the final sentence were about sperm, rather than the female reproductive tract. Faster evolution in the female reproductive tract was marginal and it was not the focus of the study.

Introduction:

4. Line 64-67. This could use clarification. I think the point is that it’s difficult to draw conclusions from previous studies on the cause of different rates of evolution in sperm compared to other features. But there was nothing erroneous about drawing the conclusion that sperm evolved slower if that’s what the results showed. In general, I think the wording in this paragraph is a bit harsh on previous studies, which may have had different aims than the present paper. The emphasis should not be on why they were wrong but on how this study builds on them.

Methods (including Supplementary Methods):

5. The analytical framework is ideal for the question and appears to have been used correctly, though no code was provided for verification. It would be useful to have a column showing the degrees of freedom for the likelihood ratio tests in Tables 1 and 2 and ESM Table 2, as these will

be different for 2-trait “pairwise” trait models versus models with > 2 traits. The authors might also include a correction for multiple likelihood ratio tests, which increases the chances of finding a significant p-value by chance (though this will be more of an issue for Table 2 than the key results in Table 1, which are far from marginal).

6. Readers would greatly benefit from a figure displaying the dated phylogenetic trees used in analyses – if not in the main paper, then at least in the supplementary materials. Important aspects of the trees are unclear from the text alone, such as the dates of key divergence events and whether the trees are fully resolved or if polytomies exist. Interested readers will certainly want to have a look at the estimated relationships among taxa, the dates of cladogenesis, and the patterns of branch lengths, all of which are easier to show than to describe verbally.

7. The authors derive age estimates from the software BEAST, which requires as input the mean and standard deviation of mutation rate in a given genetic sequence. Ideally, these parameters are based on previously published estimates of mutation rate for a locus derived from fossil or biogeographic calibration. The authors should clarify the value of these parameters used in the analysis (was it just default values?) and why they were chosen (i.e. what studies, if any, this estimate was based on).

8. I suggest the authors drop the tests of phylogenetic signal. These tests were used to assess whether traits conform to the expectations of a Brownian motion process, which is an assumption of the models used in their main analyses. But phylogenetic signal is a poor indicator of the underlying evolutionary process (see Revell et al. 2008), and the authors already conduct the more appropriate check (likelihood-based model comparison, ESM Table 2). Also, the results of the phylogenetic signal tests don’t appear to influence the traits that are eventually used in the main analysis (thorax length, which had significant phylogenetic signal, was excluded from downstream analysis, but seminal receptacle length was not).

9. Supp Mat: In the paragraph, “Drosophila and Lepidoptera Phylogenies”, the phrase “monophyletic clade” is redundant. A clade is monophyletic by definition.

10. Supp Mat: In the first paragraph under “Phylogenetic Analyses”, the topic sentence is a bit misleading. The authors don’t use a three step approach to determine whether traits conform to Brownian expectations; they use a two-step approach to determine if traits are Brownian, after which a subset of traits were used in the main evolutionary rate analysis.

11. Supp Mat: In the last sentence of the first paragraph in “Phylogenetic Analyses”, the wording suggests that data transformations were performed with the sole purpose of creating unitless variables, but I’d quickly note that an additional and very important purpose of the transformations is to bring the evolutionary change of different traits onto a common scale (i.e. to avoid the problem of comparing a 1mm change in sperm cell length to a 1mm change in wing length – changes that are proportionally very different).

12. Supp Mat: In the section “Assessing phylogenetic signal”, “Blomberg’s K” should be Blomberg’s K .

13. Sup Mat: In the first paragraph of “Assessing phylogenetic signal”, the interpretation of Blomberg K values is incorrect. K values greater than 1 indicate that traits in different lineages have diverged less (i.e. are more similar) than expected under BM.

Results

14. Some readers will be expecting to see AICc values (or, alternatively, delta AICc values) reported in tables 1 and 2, along with the results of likelihood ratio tests, as the former are more commonly encountered. Presenting the results this way should have no effect on the paper’s conclusions but may help some readers assess confidence in model comparisons.

15. The y-axes in the three panels in figure 1 should be equivalent.

Discussion:

16. In the discussion, I'd like to see the authors refer back to the studies mentioned in line 59. It would be interesting to hear why the results appear to be different in these insects than in beetles and Anolis lizards.

Refs

Obbard, DJ. et al. 2012. Estimating divergence dates and substitution rates in the *Drosophila* phylogeny. *Molecular Biology and Evolution* 29: 3459–3473.

Gao et al. 2007. Molecular phylogeny of the *Drosophila obscura* species group, with emphasis on the Old World species. *BMC Evolutionary Biology* 7: 1–12.

Mitter, C. et al. 2017. Phylogeny and evolution of Lepidoptera. *Annu. Rev. Entomol.* 62: 265–83.

Revell, LJ. et al. 2008. Phylogenetic signal, evolutionary process, and rate. *Systematic Biology* 57: 591–601.

Decision letter (RSPB-2019-2260.R0)

28-Oct-2019

I am writing to inform you that this version of your manuscript RSPB-2019-2260 entitled "Repeated evidence that the accelerated evolution of sperm is associated with their fertilization function" has, in its current form, been rejected for publication in *Proceedings B*.

This action has been taken on the advice of referees, who have recommended that substantial revisions are necessary. With this in mind we would be happy to consider a resubmission, provided the comments of the referees are fully addressed. However please note that this is not a provisional acceptance. Please note at the outset, that it is very unusual, for us to invite a 2nd round of full revision and another resubmission. The decision has been made based on the potential interest and impact of your study design and findings, though there is of course no guarantee of eventual publication in *PRSB*. There are some fundamental issues, relating to the choice and use of datasets, the nature of phylogenetic analysis, and the dynamics of insect trait evolution. It is thereby important, to both provide a full and considered response to the issues raised, as well as providing a convincing revision of the manuscript. I would say at this stage, that unless the resubmitted manuscript has a strong and favoured report, there will be no further opportunity for significant revision. It is important to recognise these points, before investing additional time.

Please find below the comments made by the referees, not including confidential reports to the Editor, which I hope you will find useful.

Sincerely,

Professor Gary Carvalho
 mailto: proceedingsb@royalsociety.org

Associate Editor Board Member

Comments to Author:

Dear Authors,

I have now received two careful and constructive reviews for your manuscript "Repeated evidence that the accelerated evolution of sperm is associated with their fertilization function". First, I would like to thank the authors for revising their manuscript and for the most part addressing all prior Reviewer concerns and comments. The restructured Introduction and Methods allow the reader to more easily and better link the motivation for the study with the subsequent analyses and data. I believe the manuscript is much improved and has the potential to be of really strong scientific importance and relevance for Proceedings B.

As with the previous submission, both Reviewers agree with me about the manuscript's potential. However, one Reviewer still has very relevant concerns and issues that need to be directly addressed: mainly regarding the phylogenetic estimations and analyses (for example the choice to still include moths). This issue partly came up in the last submission and was not directly addressed here.

It is my hope that these new comments again help in revising the manuscript and I look forward to reading the next submission.

Reviewer(s)' Comments to Author:

Referee: 2

Comments to the Author(s).

The authors have addressed all of the comments and concerns I raised in my review. My only suggestion, which they should feel free to ignore, is that they reconsider their new title and perhaps instead use their old title but with accelerated in place of rapid.

Referee: 4

Comments to the Author(s).

This manuscript presents an analysis of sperm cell evolution and provides a test of the principal hypothesis for their extreme morphology diversity. The authors ask three questions:

- 1) Does sperm cell morphology evolve quicker than that of other male traits?

- 2) If so, does the higher rate of evolution in sperm relate to their function in fertilization?
- 3) Finally, do the various components of a sperm cell evolve in concert, or do some components evolve more quickly than others?

The authors recognize a naturally-occurring test of their first two questions in species groups with heteromorphic sperm (i.e. species that produce fertile and non-fertile sperm morphs: moths, butterflies, and *Drosophila obscura*). This is a clever and original insight. By comparing morphological evolution in fertile and non-fertile sperm, which originate in the same tissue and occur in the same part of the body, the authors remove many confounding variables and isolate function as the primary predictor of evolutionary rate. The authors report faster rates of evolution in fertile sperm versus non-fertile sperm and wing length, and the find that rates of evolution in fertile sperm match those of the female reproductive tract, consistent with a role of fertilization in driving sperm evolution. The approach is innovative and the study has the potential to be impactful, but two serious issues should be addressed.

Major Comments

1. Unreliable phylogenies for Lepidopterans. The authors made a concerted effort to use the same or similar genetic sequences to estimate phylogenies in the three species groups (i.e. COII was used to construct and date trees for *Drosophila*, the related COI was used to construct and date trees for butterflies and, separately, moths), but this is a bit misguided, as the greater priority is to use sequences that allow reliable estimates of phylogenies over the timescales in question.

The three species groups used in this study represent different taxonomic ranks and have diversified over vastly different timescales – *Drosophila obscura* is a subgroup within a genus that arose ~15-20 my (Gao et al. 2007, Obbard et al. 2012), while Lepidoptera is an entire order of insects that arose ~190 my (Mitter et al. 2017) and butterflies are a superfamily within Lepidoptera that evolved ~80-140 my (Mitter et al. 2017). Mitochondrial DNA has a relatively high substitution rate, making it useful for estimating relationships among closely-related species. For the *Drosophila obscura* group, COII is likely a good choice, but COI is unreliable for estimating phylogenies among the Lepidopterans, which contain species from different families and superfamilies, due to saturation. The authors note that COI is the only gene available for the moths in their dataset. One potential option is to use published phylogenies to find the topology of species in the dataset and then fix this topology when building a tree in BEAST, thus using COI only to estimate divergence times. But even then, dates from the more deeply-diverged splits are unlikely to be accurate. A potential fix could be use 2nd and 3rd codon positions, which have lower substitution rates (see references within Obbard et al. 2012). An alternative route may be to drop the moths from the analyses.

It could be argued that this problem is unlikely to affect the key results. The phylogeny will be inaccurate, and the estimated rates of evolution will be inaccurate. But estimated rates will be inaccurate in the same way for all traits. It might not be wrong to conclude that one trait evolves faster than another.

2. A second significant issue is the splitting of Lepidopterans in the dataset into two insect groups: butterflies (a Lepidopteran clade) and moths (non-butterfly Lepidopterans). This appears to have been done because data on the female reproductive tract were only available for moths, but it creates the misleading impression that the evolutionary dynamics uncovered in this analysis (i.e. faster rates of evolution in fertile versus non-fertile sperm) are independently replicated three times in insects. Butterflies are nested within moths and did not evolve heteromorphic sperm independently, and their differential evolutionary rates should not be considered independent either. The valid approach would be to join moths and butterflies into a Lepidopteran tree for the majority of analyses, then prune out the butterflies for the test that includes the female reproductive tract. This change should be reflected in the reporting of the results.

Minor Comments

Abstract:

1. Here and throughout the paper, the term “divergence” (e.g. phenotypic divergence, evolutionary divergence, morphological divergence) is used with reference to whole species groups, and to me this creates some confusion. When I think of divergence I think of the evolution of differences between two lineages, such as sister species that have recently diverged. I recommend replacing divergence with evolution to make it clearer that we’re talking about whole species groups.
2. Line 28-30. This sentence makes it sound as though the only component of sperm morphology that was studied was head length. This more interesting result that should be summarized here is that the two types of sperm have evolved quite differently (faster head evolution in fertile sperm, faster flagellum evolution in non-fertile sperm).
3. Line 33. I think this would be stronger if the final sentence were about sperm, rather than the female reproductive tract. Faster evolution in the female reproductive tract was marginal and it was not the focus of the study.

Introduction:

4. Line 64-67. This could use clarification. I think the point is that it’s difficult to draw conclusions from previous studies on the cause of different rates of evolution in sperm compared to other features. But there was nothing erroneous about drawing the conclusion that sperm evolved slower if that’s what the results showed. In general, I think the wording in this paragraph is a bit harsh on previous studies, which may have had different aims than the present paper. The emphasis should not be on why they were wrong but on how this study builds on them.

Methods (including Supplementary Methods):

5. The analytical framework is ideal for the question and appears to have been used correctly, though no code was provided for verification. It would be useful to have a column showing the degrees of freedom for the likelihood ratio tests in Tables 1 and 2 and ESM Table 2, as these will be different for 2-trait “pairwise” trait models versus models with > 2 traits. The authors might also include a correction for multiple likelihood ratio tests, which increases the chances of finding a significant p-value by chance (though this will be more of an issue for Table 2 than the key results in Table 1, which are far from marginal).
6. Readers would greatly benefit from a figure displaying the dated phylogenetic trees used in analyses – if not in the main paper, then at least in the supplementary materials. Important aspects of the trees are unclear from the text alone, such as the dates of key divergence events and whether the trees are fully resolved or if polytomies exist. Interested readers will certainly want to have a look at the estimated relationships among taxa, the dates of cladogenesis, and the patterns of branch lengths, all of which are easier to show than to describe verbally.
7. The authors derive age estimates from the software BEAST, which requires as input the mean and standard deviation of mutation rate in a given genetic sequence. Ideally, these parameters are based on previously published estimates of mutation rate for a locus derived from fossil or biogeographic calibration. The authors should clarify the value of these parameters used in the analysis (was it just default values?) and why they were chosen (i.e. what studies, if any, this estimate was based on).
8. I suggest the authors drop the tests of phylogenetic signal. These tests were used to assess whether traits conform to the expectations of a Brownian motion process, which is an assumption

of the models used in their main analyses. But phylogenetic signal is a poor indicator of the underlying evolutionary process (see Revell et al. 2008), and the authors already conduct the more appropriate check (likelihood-based model comparison, ESM Table 2). Also, the results of the phylogenetic signal tests don't appear to influence the traits that are eventually used in the main analysis (thorax length, which had significant phylogenetic signal, was excluded from downstream analysis, but seminal receptacle length was not).

9. Supp Mat: In the paragraph, "Drosophila and Lepidoptera Phylogenies", the phrase "monophyletic clade" is redundant. A clade is monophyletic by definition.

10. Supp Mat: In the first paragraph under "Phylogenetic Analyses", the topic sentence is a bit misleading. The authors don't use a three step approach to determine whether traits conform to Brownian expectations; they use a two-step approach to determine if traits are Brownian, after which a subset of traits were used in the main evolutionary rate analysis.

11. Supp Mat: In the last sentence of the first paragraph in "Phylogenetic Analyses", the wording suggests that data transformations were performed with the sole purpose of creating unitless variables, but I'd quickly note that an additional and very important purpose of the transformations is to bring the evolutionary change of different traits onto a common scale (i.e. to avoid the problem of comparing a 1mm change in sperm cell length to a 1mm change in wing length - changes that are proportionally very different).

12. Supp Mat: In the section "Assessing phylogenetic signal", "Blomberg's K" should be Blomberg's K .

13. Sup Mat: In the first paragraph of "Assessing phylogenetic signal", the interpretation of Blomberg K values is incorrect. K values greater than 1 indicate that traits in different lineages have diverged less (i.e. are more similar) than expected under BM.

Results

14. Some readers will be expecting to see AICc values (or, alternatively, delta AICc values) reported in tables 1 and 2, along with the results of likelihood ratio tests, as the former are more commonly encountered. Presenting the results this way should have no effect on the paper's conclusions but may help some readers assess confidence in model comparisons.

15. The y-axes in the three panels in figure 1 should be equivalent.

Discussion:

16. In the discussion, I'd like to see the authors refer back to the studies in mentioned in line 59. It would be interesting to hear why the results appear to be different in these insects than in beetles and Anolis lizards.

Refs

Obbard, DJ. et al. 2012. Estimating divergence dates and substitution rates in the *Drosophila* phylogeny. *Molecular Biology and Evolution* 29: 3459–3473.

Gao et al. 2007. Molecular phylogeny of the *Drosophila obscura* species group, with emphasis on the Old World species. *BMC Evolutionary Biology* 87:

Mitter, C. et al. 2017. Phylogeny and evolution of Lepidoptera. *Annu. Rev. Entomol.* 62: 265–83.

Revell, LJ. et al. 2008. Phylogenetic signal, evolutionary process, and rate. *Systematic Biology*.

Author's Response to Decision Letter for (RSPB-2019-2260.R0)

See Appendix B.

RSPB-2020-0688.R0

Review form: Reviewer 4

Recommendation

Accept with minor revision (please list in comments)

Scientific importance: Is the manuscript an original and important contribution to its field?

Good

General interest: Is the paper of sufficient general interest?

Good

Quality of the paper: Is the overall quality of the paper suitable?

Good

Is the length of the paper justified?

Yes

Should the paper be seen by a specialist statistical reviewer?

No

Do you have any concerns about statistical analyses in this paper? If so, please specify them explicitly in your report.

Yes

It is a condition of publication that authors make their supporting data, code and materials available - either as supplementary material or hosted in an external repository. Please rate, if applicable, the supporting data on the following criteria.

Is it accessible?

Yes

Is it clear?

Yes

Is it adequate?

Yes

Do you have any ethical concerns with this paper?

No

Comments to the Author

I sincerely thank the authors for their time and effort in addressing several of the issues identified in the previous manuscript. In most respects I am satisfied with the paper in its current form. However, the Lepidopteran phylogeny stands out as an issue.

While I recognize the limitations to sequence data available in genbank and empathize in this regard, the problem remains that COII sequences probably evolve too quickly to accurately resolve relationships among the Lepidopterans included in this study. If this tree had been constructed to simply "correct" for phylogeny given some sort of regression analysis, this would be less of an issue, as several studies have shown a compromised phylogeny to be preferable to no phylogeny at all. However, in the present case, the primary results of the paper are based on estimates of evolutionary rates that are directly derived from estimates of branch lengths in the phylogeny. Confidence in evolutionary rate estimates are thus limited by low confidence in branch length estimates derived from COII. Presented as is, I think readers would view these Lepidopteran results with strong suspicion. I have three suggestions the authors may consider.

1. Re-run the Lepidopteran analysis using the most comprehensive Lepidopteran tree available in the literature.

- Based on my very quick search, this would be the Kawahara et al. (2019 PNAS) tree, which is based on a huge genomic and transcriptomic dataset. This tree contains 31 of the genera used in the present analysis, so the Lepidopteran test could easily be re-run in a short period of time using this published tree (pruned to the 31 relevant tips). The authors could decide to restrict their analysis to just these 31 lineages, or, if such an analysis were to give them very similar results to what they report here, they could use it to say that their larger results are unaffected by using just COII.

2. Remove Lepidoptera from the analysis.

- In my opinion, the results from *Drosophila* alone are sufficient to substantiate each of the key findings in this paper (i.e. faster rates of evolution in fertile versus non-fertile sperm, matching rates of evolution in fertile sperm and female reproductive organs, and differential rates of evolution in different sperm components), all of which are interesting. The manuscript would be relatively unchanged by focusing on just *Drosophila* (with the exception of the title).

3. Demonstrate that the issue with COII is unimportant.

- The authors could provide a simple saturation plot to demonstrate that the COII sequence data has sufficient "depth" to estimate the rates reported here. If the plot shows little or no saturation, then the tree and the rate estimates would be substantiated.

I think that none of the above suggestions would dramatically alter the current paper or take much time, but they would alleviate the main methodological concern that readers would probably have from the Lepidopteran results in their current form.

Minor comments:

1. Line 54-55: wording here is confusing. Would suggest something shorter along the lines of "however, in these studies, rates of sperm diversification were compared ..."

2. Line 260: mixed-up wording. the "and" should be moved from its position to go between "organ length" and "sperm length"

3. I would prefer to see the paragraph on phylogenetic tree construction in the results section itself, rather than in supplementary methods. It doesn't seem particularly long and readers could more directly assess confidence in results without having to dig into online supplements.

4. Wording regarding the main test of the paper is confusing in two areas.

Decision letter (RSPB-2020-0688.R0)

22-Apr-2020

I am writing to inform you that this version of your manuscript RSPB-2020-0688 entitled "Repeated evidence that the accelerated evolution of sperm is associated with their fertilization function" has, in its current form, been rejected for publication in Proceedings B.

This action has been taken on the advice of referees, who have recommended that substantial revisions are necessary. With this in mind we would be happy to consider a resubmission, provided the comments of the referees are fully addressed. However please note that this is not a provisional acceptance.

Please find below the comments made by the referees, not including confidential reports to the Editor, which I hope you will find useful.

Sincerely,
Professor Gary Carvalho
<mailto:proceedingsb@royalsociety.org>

Associate Editor Board Member

Comments to Author:

We have now received back comments from a statistical reviewer for the manuscript "Repeated evidence that the accelerated evolution of sperm is associated with their fertilization function". I think the main positive takeaway is that the reviewer feels the overall main results of the paper, especially as they pertain to the *Drosophila* phylogeny, are compelling. I wholeheartedly agree. However, I have shared their and others concern that the Lepidopteran results are less strong, and instead take away from the overall scope of the manuscript as written (or analyzed). There have now been a few attempts to fix these issues but, as the Reviewer nicely points out in their comments, glaring ones still remain. If the authors are not able to directly address the comments of the Reviewer they may find more success sending their paper elsewhere. If they are able to address these comments we would more than welcome the resubmission.

Reviewer(s)' Comments to Author:

Referee: 4

Comments to the Author(s).

I sincerely thank the authors for their time and effort in addressing several of the issues identified in the previous manuscript. In most respects I am satisfied with the paper in its current form. However, the Lepidopteran phylogeny stands out as an issue.

While I recognize the limitations to sequence data available in genbank and empathize in this regard, the problem remains that COII sequences probably evolve too quickly to accurately resolve relationships among the Lepidopterans included in this study. If this tree had been constructed to simply "correct" for phylogeny given some sort of regression analysis, this would be less of an issue, as several studies have shown a compromised phylogeny to be preferable to no phylogeny at all. However, in the present case, the primary results of the paper are based on estimates of evolutionary rates that are directly derived from estimates of branch lengths in the phylogeny. Confidence in evolutionary rate estimates are thus limited by low confidence in branch length estimates derived from COII. Presented as is, I think readers would view these Lepidopteran results with strong suspicion. I have three suggestions the authors may consider.

1. Re-run the Lepidopteran analysis using the most comprehensive Lepidopteran tree available in the literature.

- Based on my very quick search, this would be the Kawahara et al. (2019 PNAS) tree, which is based on a huge genomic and transcriptomic dataset. This tree contains 31 of the genera used in the present analysis, so the Lepidopteran test could easily be re-run in a short period of time using this published tree (pruned to the 31 relevant tips). The authors could decide to restrict their analysis to just these 31 lineages, or, if such an analysis were to give them very similar results to what they report here, they could use it to say that their larger results are unaffected by using just COII.

2. Remove Lepidoptera from the analysis.

- In my opinion, the results from *Drosophila* alone are sufficient to substantiate each of the key findings in this paper (i.e. faster rates of evolution in fertile versus non-fertile sperm, matching rates of evolution in fertile sperm and female reproductive organs, and differential rates of evolution in different sperm components), all of which are interesting. The manuscript would be relatively unchanged by focusing on just *Drosophila* (with the exception of the title).

3. Demonstrate that the issue with COII is unimportant.

- The authors could provide a simple saturation plot to demonstrate that the COII sequence data has sufficient "depth" to estimate the rates reported here. If the plot shows little or no saturation, then the tree and the rate estimates would be substantiated.

I think that none of the above suggestions would dramatically alter the current paper or take much time, but they would alleviate the main methodological concern that readers would probably have from the Lepidopteran results in their current form.

Minor comments:

1. Line 54-55: wording here is confusing. Would suggest something shorter along the lines of "however, in these studies, rates of sperm diversification were compared ..."

2. Line 260: mixed-up wording. the "and" should be moved from its position to go between "organ length" and "sperm length"

3. I would prefer to see the paragraph on phylogenetic tree construction in the results section itself, rather than in supplementary methods. It doesn't seem particularly long and readers could more directly assess confidence in results without having to dig into online supplements.

4. Wording regarding the main test of the paper is confusing in two areas.

Author's Response to Decision Letter for (RSPB-2020-0688.R0)

See Appendix C.

RSPB-2020-1286.R0

Review form: Reviewer 4

Recommendation

Accept with minor revision (please list in comments)

Scientific importance: Is the manuscript an original and important contribution to its field?

Excellent

General interest: Is the paper of sufficient general interest?

Excellent

Quality of the paper: Is the overall quality of the paper suitable?

Good

Is the length of the paper justified?

Yes

Should the paper be seen by a specialist statistical reviewer?

No

Do you have any concerns about statistical analyses in this paper? If so, please specify them explicitly in your report.

No

It is a condition of publication that authors make their supporting data, code and materials available - either as supplementary material or hosted in an external repository. Please rate, if applicable, the supporting data on the following criteria.

Is it accessible?

Yes

Is it clear?

Yes

Is it adequate?

Yes

Do you have any ethical concerns with this paper?

No

Comments to the Author

I thank the authors for their attention to previous concerns regarding the Lepidopteran phylogeny.

The main issue with the previous manuscript was that the authors inferred evolutionary rates for sperm traits based on a Lepidopteran phylogeny constructed from COII, a mitochondrial gene subunit with a substitution rate that is probably too high to accurately infer relationships among such anciently-diverged lineages. I had no major criticisms of the *Drosophila* results, which were and are sufficient to substantiate the paper's main findings.

In the revised manuscript the authors employ a recently-published Lepidopteran phylogeny based on extensive genomic and transcriptomic datasets to estimate rates of evolution in sperm traits. Branch lengths in this phylogeny are much more reliable than the previous tree based on COII, and as a consequence the estimates of evolutionary rate are also much more reliable. The cost to this improved approach is that only 12 of the 135 Lepidopteran species for which the authors obtained trait data could be included in the revised analysis. Nevertheless, the analysis found quite strong support for a more complex model of trait evolution (i.e. a model in which different traits had different evolutionary rates). I am therefore satisfied with these newer results.

One final and minor issue concerns the scope of inference for Lepidoptera. I think the authors need to be clearer in the abstract and in the discussion that their results, while interesting and novel, are true for *Drosophila obscura* and a small group of Lepidopterans for which data were available, not necessarily for Lepidoptera as a whole. For instance, the authors write "We provide consistent evidence that fertile sperm length evolves faster than non-fertile sperm length in *Drosophila* from the *obscura* group and Lepidoptera, two groups with heteromorphic sperm." To me the Lepidopteran part of this sentence is going a bit far, as the inference is based on 12 species and Lepidoptera contains ~180k described species. A slight wording change to something along the lines of "We provide consistent evidence that fertile sperm length evolves faster than non-fertile sperm length in *Drosophila* from the *obscura* group and in a group of Lepidopterans for which sperm trait data were available ..." would clear that up and leave me no further objections.

Decision letter (RSPB-2020-1286.R0)

07-Jul-2020

Dear Dr Fitzpatrick:

Your manuscript has now been peer reviewed and the reviews have been assessed by an Associate Editor. The reviewers' comments (not including confidential comments to the Editor) and the comments from the Associate Editor are included at the end of this email for your reference. As you will see, the reviewers and the Editors have raised some concerns with your manuscript and we would like to invite you to revise your manuscript to address them.

Research ethics:

Use of animals and field studies:

All supplementary materials accompanying an accepted article will be treated as in their final form. They will be published alongside the paper on the journal website and posted on the online figshare repository. Files on figshare will be made available approximately one week before the

accompanying article so that the supplementary material can be attributed a unique DOI. Please try to submit all supplementary material as a single file.

Please submit a copy of your revised paper within three weeks. If we do not hear from you within this time your manuscript will be rejected. If you are unable to meet this deadline please let us know as soon as possible, as we may be able to grant a short extension.

Best wishes,
Professor Gary Carvalho
mailto:proceedingsb@royalsociety.org

Associate Editor Board Member

Comments to Author:

Dear Authors,

Your paper has now been reassessed by a past reviewer and myself. Both of us find the manuscript much improved once again, and commend the authors on a fine job of answering all the reviewer queries and comments. The reviewer has one final important comment that should be addressed by the authors. Namely, they would like the authors to specifically address the obvious limitations in scope for the Lepidopteran data. I agree that this is a much needed step, and one that will not take away from the main and novel inferences that has been made from these data sets. I look forward to the revised manuscript.

Reviewer(s)' Comments to Author:

Referee: 4

Comments to the Author(s).

I thank the authors for their attention to previous concerns regarding the Lepidopteran phylogeny.

The main issue with the previous manuscript was that the authors inferred evolutionary rates for sperm traits based on a Lepidopteran phylogeny constructed from COII, a mitochondrial gene subunit with a substitution rate that is probably too high to accurately infer relationships among such anciently-diverged lineages. I had no major criticisms of the Drosophila results, which were and are sufficient to substantiate the paper's main findings.

In the revised manuscript the authors employ a recently-published Lepidopteran phylogeny based on extensive genomic and transcriptomic datasets to estimate rates of evolution in sperm traits. Branch lengths in this phylogeny are much more reliable than the previous tree based on COII, and as a consequence the estimates of evolutionary rate are also much more reliable. The cost to this improved approach is that only 12 of the 135 Lepidopteran species for which the authors obtained trait data could be included in the revised analysis. Nevertheless, the analysis found quite strong support for a more complex model of trait evolution (i.e. a model in which different traits had different evolutionary rates). I am therefore satisfied with these newer results.

One final and minor issue concerns the scope of inference for Lepidoptera. I think the authors need to be clearer in the abstract and in the discussion that their results, while interesting and novel, are true for *Drosophila obscura* and a small group of Lepidopterans for which data were available, not necessarily for Lepidoptera as a whole. For instance, the authors write "We provide consistent evidence that fertile sperm length evolves faster than non-fertile sperm length in *Drosophila* from the *obscura* group and Lepidoptera, two groups with heteromorphic sperm." To me the Lepidopteran part of this sentence is going a bit far, as the inference is based on 12 species and Lepidoptera contains ~180k described species. A slight wording change to something along the lines of "We provide consistent evidence that fertile sperm length evolves faster than non-fertile sperm length in *Drosophila* from the *obscura* group and in a group of Lepidopterans for which sperm trait data were available ..." would clear that up and leave me no further objections.

Author's Response to Decision Letter for (RSPB-2020-1286.R0)

See Appendix D.

Decision letter (RSPB-2020-1286.R1)

15-Jul-2020

Dear Dr Fitzpatrick

I am pleased to inform you that your manuscript entitled "Repeated evidence that the accelerated evolution of sperm is associated with their fertilization function" has been accepted for publication in Proceedings B.

Open Access

Your article has been estimated as being 8 pages long. Our Production Office will be able to confirm the exact length at proof stage.

Paper charges

Sincerely,

Professor Gary Carvalho
Editor, Proceedings B
mailto: proceedingsb@royalsociety.org

Associate Editor:
Board Member
Comments to Author:
Dear Authors,

I want to thank you for your timely response to all comments. This is an impressive manuscript.

Appendix A

01-Aug-2019

Dear Dr Fitzpatrick:

I am writing to inform you that your manuscript RSPB-2019-1477 entitled "Convergent evidence for the rapid evolution of sperm morphology" has, in its current form, been rejected for publication in Proceedings B.

This action has been taken on the advice of referees, who have recommended that substantial revisions are necessary. With this in mind we would be happy to consider a resubmission, provided the comments of the referees are fully addressed. However please note that this is not a provisional acceptance.

Sincerely,

Professor Gary Carvalho
mailto: proceedingsb@royalsociety.org

Associate Editor
Board Member: 1
Comments to Author:
Dear Authors,

I have now received three careful and constructive reviews for your manuscript "Convergent evidence for the rapid evolution of sperm morphology". All reviewers see merit in the topic and feel that the manuscript can be of broad interest to the general audience at Proc B. However they each also raise many issues that need to be addressed before that decision can be made. For example, although the

concept and experimental design of the experiment are strong, the authors should simplify the result tables, better justify the analyses used and parameters chosen (as they pertain to the main objectives: see comments by Reviewer 1) and provide better descriptions and explanations of the results (i.e. impact, why are the main findings important?). They should also revise the framing of the introduction (see major comments by Reviewer 2), and be careful and consistent with their sperm-related terminology. I hope that you find the comments of the reviewers helpful in revising your manuscript.

Summary response: We have taken all reviewer comments on board: we simplify the results table, move information from the ESM into the main text that helps to more easily see (justify) the connection between motivation for the study, the parameters chosen, and the analyses used, and by reframing the Introduction we also improve the readability of the impact and why main findings are important. We have ensured consistent use of sperm-related terminology, including fertile vs non-fertile sperm and clarified measurements were of length. We thank the reviewers for their appreciation of the study and subsequent helpful comments that improved the manuscript.

Reviewer(s)' Comments to Author:

Referee: 1

Comments to the Author(s)
Reviewer Comments

In this comparative study the authors provide compelling evidence for the similarity in evolutionary rates of sperm morphology in two sperm heteromorphic insect groups, which are in turn linked to sperm function (fertilizing or not). In addition, the authors compare these evolutionary rates of sperm trait diversification with those of other key reproductive and somatic traits. The paper reads well, the statistical analyses sound, and the study's obvious importance to the field of evolutionary biology is well described and supported. However, at times it lacks clarity/important details for some analytical choices and in the presentation of the, necessarily complex, statistical results. Below, I list the main issues that should be addressed.

Main issues

- It is important to briefly list (e.g. in Intro L119) and later (e.g. in Methods L164) adequately describe and, very importantly, justify the choices of all the non-sperm traits used in the analyses. In particular, the choice of somatic traits. For instance, is Lepidopteran forewing length also restricted to males like the thorax length? Are these somatic traits chosen for their negligible or important role in (pre-copulatory) sexual selection? The reader should not have to read the ESM to get this important methodological information. I assume this was due to word limits. You can perhaps move some more of the technical details (i.e. explaining Ref #46) to Supplementary and return these key biologically relevant aspects of the Methods to the main text.

Response: Thanks for this suggestion. We have now moved text from the ESM to the main text and added information justifying the choice of traits in our analyses (lines 146-167). To make comparisons among groups more straightforward we also decided to use wing length in *Drosophila* as a point of comparison to forewing length in butterflies and moths. In keeping with the suggestion by the reviewer, we have moved technical details to the ESM.

- I was wondering the need to include the butterflies in this study, since you have no data on female reproductive structure size. Perhaps include only Moths for the main paper and have the Lepidoptera and Butterfly analyses as supplementary material? I assume that there was a reason to compare evolutionary rates of sperm morphology with reproductively meaningful sex-specific traits in both sexes (based on the details in ESM Table S2). If so, the story is rather incomplete for butterflies and Moths make a strong case by themselves for ‘convergent evolution’ with the *Drosophila* group.

Response: We appreciate where the reviewer is coming from but have elected to keep the analyses on butterflies and moths in the manuscript. Here’s why: a key part of our analyses is testing the evolutionary rate of sperm morphological diversity while controlling for confounding factors present in other studies (i.e., traits arising from different tissues or acting during pre – vs post – copulatory sexual selection). Therefore, we tested this in as many different sperm heteromorphic groups as possible. Despite the lack of data on the female reproductive tract, the analyses with butterflies demonstrate a remarkably consistent pattern to those observed in moths and *Drosophila*. Moreover, as we detail below, we incorrectly remarked that moths and butterflies are both monophyletic groups. This is not the case. Butterflies are monophyletic but moths are paraphyletic. Therefore, we have kept the two groups separate in our analyses and have instead removed the Lepidoptera analyses all together.

- Tables 1 and 2 are unduly complex. Perhaps for each combination of 2 (or 4) traits, the evolutionary rates should be in the same column instead of a 2x2 (or 4x4) matrix where the top off-diagonal is left blank because it is a repeat of the bottom off-diagonal. It would lengthen the table, yes, but into a more portrait cf. landscape format. I also suggest placing the relevant ‘pairwise analyses’ to the right of the respective evolutionary rate estimation models outputs. It will make it easier to follow the results section and the plots.

Response: Another great point. We have dramatically simplified the tables. While we didn’t follow all of the reviewer’s useful suggestions we took the spirit of the comment on board and have made the table much easier to follow. We have also placed all the pairwise analysis information into the ESM as this information was already presented using superscript letters in Figure 1.

Other issues

L1: the title should stress that we are working on sperm heteromorphic groups. I’m not sure ‘rapid’ is warranted

Response: We have changed the title of our manuscript. We agree that ‘rapid’ may not be warranted. However, we disagree with specifying that we are focusing on sperm heteromorphic groups in the title as we argue that these groups are useful for illuminating broader patterns. Therefore, we prefer to emphasize a broader message in our title, while specifying the model groups in the abstract.

L22: ‘two taxa’ makes me think of species. Perhaps ‘two (insect) groups’ is better.

Response: Done.

L110: add a line to briefly explain what the different functions of the two sperm types are (i.e. what the non-fertile sperm is for)

Response: Done.

L119: you say here the somatic traits chosen are a proxy for naturally selected traits sensu Ref#5. Are the sizes of female reproductive tract component also such traits? Or were you referring to (male?) wing and thorax. Be more explicit.

Response: Done. We have clarified our writing here (line 158-164) and throughout the manuscript.

L164: here you list 'sperm', 'reproductive' and 'somatic' type of trait for which you will calculate evolutionary rates. Now we have three types. But as mentioned earlier, there is no detailed description and justification for all the ones used in the analyses

Response: We have added this justification to the Introduction (lines 96-129).

L202: what are these taxonomic groups? Spell them out.

Response: Done.

L207: I think that once the evolutionary rates are in the same column vs. in a matrix format in the Table, it will make it easier to see how one is 'slower' or 'faster' and by how much.

Response: We have included the observed rates side by side to the common rate (which is now in brackets) to make it easier to compare these two models. The observed evolutionary rates are now presented in bold for each model to make it easier to directly compare values.

L233&L298: don't forget to add the measure of the trait e.g. 'receptacle length'. It is important because mass or volume would make the reader think of sperm numbers being more relevant than sperm length, for instance.

Response: Done.

L234-L239: Why are these results in the text cf. Table?

Response: They have now been moved to the revised Table 2.

L292-293: How does male body size (or the other 'somatic' traits in the study) 'play a direct role in securing male fertility'. The latter should have been made clearer at the point that the choice of traits is justified.

Response: We were referring to fertile sperm here, not male body size. We have now clarified this sentence (line 260-262).

Table 1 (&2): Are the pairwise comparisons using fertile & non-fertile sperm referent to total sperm length in all cases and for all taxonomic groups? Perhaps be clearer in the Table 2:

Response: We have made this explicit in the table by indicating that we are referring to total length for these analyses. In the revised ESM where we report the pairwise analyses we also make it clear that we are referring to total length.

What is the superscript 'a' referent to in Moths and Butterflies?

Response: Thanks for picking this up. This should not have been there and has now been corrected.

Referee: 2

Comments to the Author(s)

Major comments:

This paper examines the hypothesis that fertile sperm traits evolve more quickly than other male traits (because of their function in fertilization and therefore effect on male reproductive success). To examine this hypothesis in a powerful way, the authors compile comparative data on sperm (total or head and flagellum) length, female storage organ length, and body (either thorax or forewing) length in two groups of insects (*Drosophila* spp. from the *obscura* group and *Lepidoptera*). Their analyses then compare the rates of trait evolution for fertile sperm length, non-fertile sperm length, female reproductive tract length and somatic length to ask whether fertile sperm length evolves faster than other reproductive and somatic lengths. The authors argue that by comparing how these two sperm types evolve – since they are generated in the same tissues and experience the same environment during selection- allows a powerful examination of this hypothesis that sperm traits evolve faster than other male traits.

I think the analyses presented in this paper are powerful and interesting. I love the idea of comparing the evolution of sperm traits for those sperm involved directly in fertilization (i.e. eusperm or eupyrene sperm) and those not involved directly in fertilization (i.e. parasperm or apyrene sperm). The comparison with the rate of female reproductive tract evolution as well as somatic length trait is also interesting, timely and informative. I am therefore very much in support of publishing this paper and the analyses it reports.

There are, however, a few things about the framing of the paper that I feel need improvement for greater clarity and precision.

1) The comparison of the rate of evolution of the two sperm types really focuses on comparing sperm that are involved in fertilization and those that are not. But the framing language is about sperm traits and other male traits. Since both are sperm, but only one type functions in the typical way we think about sperm, this gets a bit confusing. The paper should be clear that the comparison is really about traits involved directly in fertilization or not. And acknowledge that the non-fertile sperm are still likely involved indirectly in fertilization and therefore also experiencing post-copulatory sexual selection. The authors could also make the impact of their work greater, if they help the reader understand why comparing fertile and non-fertile sperm types is different than comparing fertile sperm with other traits directly related to differential fertilization success and the outcome of sperm competition (e.g. testes size, other ejaculate characteristics, etc.).

Response: We have restructured the Introduction to make these points more obvious and clear as the motivation for our work (lines 96-140).

2) In my opinion, the real power of the analyses is in looking at the correlated evolution of the two

sperm types, along with the female reproductive tract and a “comparable” somatic length variable. Yet, the female reproductive tract analyses seem like a bit of a “throwaway” in the abstract and introduction as written.

Response: We agree and so have restructured the Abstract and Introduction to better emphasize the key role of the female reproductive tract in shaping sperm evolution which further highlights what we agree is a main take home message from our work.

3) There are a variety of reasons that two traits might evolve at different rates. This is not discussed in the introduction or discussion. And the discussion makes inferences about the strength/form of selection on the traits based on differences in the rate of trait evolution without acknowledging the assumptions being made. The rate of trait evolution could also easily differ if these traits have different heritability, genetic correlations, evolutionary constraints, or trade-offs. These other reasons that the rates could differ must be discussed explicitly and the assumptions being made must be acknowledged and fully transparent in the introduction and discussion.

Response: Another good point. We have emphasized throughout the Introduction and Discussion that sperm trait evolution can be influenced by a range of factors.

Minor comments:

The “readability” of the paper would benefit from you using the terms fertilizing and non-fertilizing sperm or fertile and non-fertile sperm throughout (the more precise terms of eusperm/parasperm or eupyrene/apyrene sperm could also be included in parentheses where useful). This would especially be nice to have in the Figures and Tables to remind the reader which is which.

Response: Good point. We now use fertile and non-fertile sperm throughout and specify that these refer to eusperm/eupyrene or parasperm/apyrene sperm, respectively.

Abstract:

Line 29-30: “exhibit contrasting rates of divergence.” This phrasing felt awkward to me. Perhaps state more directly “differ in their rates of divergence”

Response: Done.

Line 32-33: An admittedly picky point, but your data can’t say anything directly about whether sperm traits evolved “because of their functional role in securing fertility”. You could say we provide direct evidence that sperm length evolves more rapidly in fertile sperm, likely because of....

Response: Done.

Line 34-35: An interesting result and yet this is the first time you mention female trait evolution and so it feels like it comes out of nowhere.

Response: We have restructured the Abstract (and later the Introduction) to emphasize the female reproductive tract earlier.

Introduction:

Lines 40-45: I believe I understand what you mean here but had to think about it a bit. And the challenge stems from many things- not just the one you mention. I suggest you rephrase this.

Response: Done.

Lines 52-55: This is a really key point for your paper. I suggest expanding this explanation a bit more for the reader to explain what is confounding the interpretation of these rates of evolution. I am not at all disagreeing with your statement- just suggesting that it is a key argument that warrants further explanation since the importance of this contribution depends on the reader understanding this point.

Response: Done.

Line 80-81 Why? I do not disagree- instead I suggest you make the argument explicitly, rather than just state that it is so. Why are sperm traits expected to evolve faster than other sexually selected traits? Is it because sexual selection is stronger on them (because they have a larger effect on fitness), because they are somehow more evolvable (less constrained genetically or otherwise), have few or weaker natural selection/sexual selection tradeoffs, or have higher heritability/underlying genetic variation, or because of the combination of these things?

Response: Sperm traits, like other sexually selected traits, are expected to evolve faster than traits that are shaped primarily by natural selection. As the reviewer suggested, there are multiple factors that can influence the rate of evolutionary change, and there is no *a priori* reason to expect sperm traits to evolve faster than other sexually selected traits. We have clarified this in the text (line 50-53).

You may want to be explicit that you use natural selection to capture all of other forms of selection- e.g. viability and fecundity-since there is some disagreement about whether sexual selection is a type of natural selection or a distinct form of selection- which is irrelevant to your claim but could confuse or annoy some readers.

Response: We definitely do not want to annoy readers! Therefore, we have moved away from emphasizing the dichotomy between natural and sexual selection and now focus on the relative strength of sexual selection among the traits we examine – reproductive traits are more strongly influenced by sexual selection while wing size is more strongly influenced by natural selection (line 161-164).

Line 111: See my general comment above. I think we need a bit more clarification why this is better than past analyses on multiple sexually selected male traits.

Response: Done.

Lines 117-120: I think this is the key strength of this work and yet this is the first the reader hears about it and why female reproductive trait evolution is worth including.

Response: We have restructured the Introduction and now describe the key role of the female reproductive tract sooner in the text.

Methods:

I suggest including more concrete information about the actual variables included in the analyses within the main methods section of the paper. I felt there was insufficient information on how the other reproductive and somatic traits were chosen and how the data on them were assembled. This all made much more sense once I read the supplemental materials (and could see that only length was compared for various traits).

Response: Done. We have moved the relevant information from the ESM and expanded on it in the revised methods.

Given that both groups are monophyletic, please explain why you split the Lepidoptera into three groups? Were the results qualitatively the same if you treated Lepidoptera as one group?

Response: We could have been clearer in our language in the initial submission. We separated moths and butterflies because only butterflies are monophyletic. Moths are paraphyletic. To avoid confusion, we state this directly in the revised manuscript (lines 103-105) and have removed the Lepidoptera analyses (which simply combined moths and butterflies into a single analysis) from the manuscript.

This section would benefit from having stated objectives and explanation of how the data and analyses relate to those objectives.

Response: We have restructured the Introduction and Methods, in response to both reviewers, to better link our motivation for the study and to link the analyses and data with the main objectives of our work.

Lines 135-142 (esp. 141-142)- this is a very clear explanation of the comparison and its power. I suggest finding a place to make the comparison this clear (though in more general form) in the introduction- especially that the comparison is really otherwise same or extremely similar traits and environments but only one of which serves to fertilize the egg.

Response: Good point. We have moved this information from the ESM into the Introduction to better highlight the power of our approach.

Line 164-165 (and in general) I found the reference to "sperm morphology", "reproductive traits" "somatic traits" confusing and misleading given that only a few specific length traits are in the dataset. Why not refer specifically to sperm length, female sperm storage organ length and body length traits- or something like that- to be clear what traits are being compared.

Response: We agree. We now use the terms sperm length, female reproductive organ length, and wing length. Note that we did not use the term 'female sperm storage organ length' as only the seminal receptacle, not the spermathecal duct, acts as a sperm storage organ. We also simplified our analyses by including only wing length as a proxy measure for male body size in all our analyses.

The comparative methods used in this paper are not my area of expertise, especially the methods for comparing evolutionary rates. I found the description of the methods used clear and relatively straightforward, though readers may not all find the sigma square parameter intuitive.

Response: Thanks. We now specify what sigma square means in more simplified language (line 175-176).

Results

Line 211: It would be more correct to say “sperm length evolves faster” not “sperm evolve faster” (here and throughout the paper). This is also true for length of the sperm storage organ, etc.

Response: Done.

Line 221: In the next few lines, the observed patterns are clear, but I wondered what “opposing evolutionary rates” meant- and my initial assumption was that you meant differences in the direction of trait evolution rather than a difference in the rate of evolution.

Response: We have rephrased this sentence (lines 201-203).

Line 227- That is a truly dramatic difference in trait evolution between fertile and non-fertile sperm length.

Response: We agree!

Line 231: Like with “opposing” I found “evolve at distinct rates” a bit confusing. Do you mean the magnitude of the evolutionary rate parameters were statistically different for the same traits? If so, it seems more correct to say the evolutionary rates differed significantly.

Response: We have rephrased this sentence (lines 212-213).

Discussion:

Line 273 and throughout the paper- What does it mean to say the patterns are convergent? I am more familiar with this term meaning the independent origin of the same traits in different groups rather than similar rates of evolution. I therefore don't think convergent is the best word to capture your results.

Response: Good point. Here and throughout the manuscript we have avoided the term convergence and instead stressed the consistency of the patterns.

Line 277- I suggest writing “different rates of trait evolution instead of “rates of phenotypic diversity across traits” (or otherwise rewriting this statement) – since I don't really understand what that phrase is saying as written.

Response: Done.

Line 281- I do not agree that you can say this about “alternative selection pressures” based on your data – at least not so strongly. They evolve at different rates and there are a variety of other reasons that might happen (e.g. different heritabilities, genetic correlations, trade-offs, different strength of same selection pressure, evolutionary constraints, etc.).

Response: Good point. We have removed this phrase.

Line 289- Lines 282-290 are a powerful explanation of the results and their relevance, but- and this is an admittedly picky point- I find “dramatically different rates of phenotypic diversification” overstated given that some differences are large but others are small.

Response: We have removed this statement from the text.

Lines 316-319- These patterns are very interesting and in general that the evolutionary pattern of sperm and female tracts is one of the key strengths of this set of analyses. I suggest introducing this /setting this up a more in the introduction.

Response: Great idea! Done.

Line 328 and throughout the discussion- there are various key assumptions being made (e.g. same heritability/genetic covariances among traits) that needs to be acknowledged when you make any inference about what the rate of trait evolution says about selection pressures. I find this interpretation over-stated given that there are many other possible reasons besides the strength of selection for differences in trait evolution.

Response: Good point. We now acknowledge alternative factors that could influence the results. However, we stress in the discussion that regardless of the mechanism underlying our results the upshot is that fertile sperm evolve faster than non-fertile sperm.

Line 373- As reviewed in the introduction, a general explanation exists but has been hard to test.

Response: For clarity this text has now been omitted.

Tables 1 and 2- Great information. But it is hard to see what the significant statistical differences “mean” about the relative absolute rate and direction of evolution. The summary at the bottom is very useful. I wonder, however, if the direction of difference could be given. For example, replace the versus with greater than, less than or approx. equal symbols (or words if preferred).

Response: A great suggestion. We have dramatically restructured our tables in line with your comments and those from the other reviewers. We have followed your useful suggestion about how to structure the pairwise analyses. However, we moved this information to the ESM as the same information is presented using superscript letters in Figure 1.

Referee: 3

Comments to the Author(s)

The authors report on their test of the idea that sperm evolve faster than other male traits, by comparing the rates of phenotypic divergence of non fertile and fertile sperm in two taxa. Their argument is that fertile and non fertile sperm are produced by the same tissue, and therefore are in fact more comparable, than examining traits that are not produced by the same tissue, or encounter the same selective environment. They also compare rates of evolution of sperm components, and female

reproductive tracts.

In line with their predictions, they find that fertile sperm evolve faster than non fertile sperm, and faster than body size, but at the same rate than female reproductive tract. They also find that the head evolves faster in the fertile sperm, while the flagellum evolves faster in non fertile sperm.

I like the motivation of the paper, and there are interesting repercussions for proposing this novel way of examining sperm evolution. The paper is generally well written (some minor suggestions follow), but clarification is needed in places. I am not familiar with the analysis, so I cannot comment as to whether they have been deployed properly, so having a reviewer who knows these methods would be important.

Line 47: change "is" for "are"

Response: Done.

Line 48: insert comma after sexual selection

Response: Done.

Lines 52-55: can you clarify the different selective episodes you are referring to here?

Response: Done.

Line 72: after per se, please elaborate why. Perhaps providing an example would be useful.

Response: Done.

Line 83-85: I don't think this is true anymore. Perhaps simply starting this paragraph with the next sentence would avoid the need for this generalization.

Response: Done.

Lines 145-147: A recent paper by Sakai et al 2019 in PNAS showed that parasperm in *B. mori* (lepidoptera) is necessary for sperm migration to the female organs, offering strong evidence that parasperm may not be cheap fillers. These data I think may change how you interpret some of your results in the discussion.

Response: Great! Thanks for pointing this out. We now include this useful paper in the text.

Line 167: Insert sample sizes here please. These are unclear in the results. How many individuals were measured to obtain estimates per species? are these averages? Also insert reduced sample sizes at the end of line 169.

Response: We have restructured the Methods and now refer to the overall sample sizes (line 146-148), while highlighting that the sample sizes for specific analyses are presented in Tables 1-2 (line 167).

Line 176-181: I suggest deleting this last few sentences since the next section explains this again. Add

“Supplementary methods” to the parenthesis that says ESM.

Response: Done.

Line 228-229: replace “was on the border of statistical significance” with the actual value.

Response: Done.

Line 233: What measurements of the seminal receptacle did you analyze? Length? volume? In the figure it says length so please clarify if that is it. The spermathecal duct and the seminal receptacle have different functions in these two taxa. The seminal receptacle stores sperm, whereas the duct is the passageway to the spermatheca, so how does the different function of these organs affect your conclusions?

Response: Done. We now clarify what measures were analysed (lines 150-158). Indeed, the seminal receptacle and spermatheca have different functions in these insect groups. However, we aren't able to comment on how the different roles of these organs might influence our results. The key similarity is that these female reproductive organs are correlated with sperm traits in the groups we are examining.

Line 246: Why does forewing length evolve faster than thorax length? Do you have data for thorax length in lepidoptera that would be more comparable? forewing length may be under sexual selection.

Response: Forewing length does not evolve faster than thorax length. For each of the observed evolutionary rates you can't compare across phylogenetic trees. In other words, the observed rates are only comparable within each analysis. Since we didn't directly compare wing length with thorax length we can't say anything about the relative rate of evolution of these traits. To avoid confusion we have removed analyses using thorax length as a measure of body size for *Drosophila* and instead use wing length.

Line 268: Here it says the rate of evolution of forewing length is the same as infertile sperm, but in line 246, it says that forewing length evolve at an intermediate rate between fertile and infertile sperm. Please clarify that you are comparing moths vs. butterflies in the text.

Response: Thanks for catching this. We have corrected the text (lines 237-241).

Line 301-319: This needs clarification. “Female reproductive tract” is not interchangeable with spermathecal duct length or seminal receptacle length. The female reproductive tract has many components that have different functions in addition to interacting with male reproductive components, and even those interactions can be quite diverse: Interacting with the spermatophore, the aedagus, or the ejaculate, so clarifying the role you are referring to when it comes to sperm interactions would seem important here. This section needs rewriting.

At the end, again I think it is important to consider the new evidence provided by Sakait et al 2019 I reconsidering what the role of infertile sperm may be in lepidoptera.

Response: Done.

Line 352-354. This requires more explanation. Are you suggesting that sperm fit just right in the female

tract so if they were longer they would not fit?

Response: We were referring to a genetic covariance between sperm length and female reproductive tract length. We now clarify this in the text (line 328).

Line 357-361: Cool idea!

Response: Thanks!

You might want to cite Higginson and Pitnick (2011) for a summary of the possible functions of parasperm.

Response: Done.

Appendix B

Dear Professor Carvalho and Associate Editor Board Member,

Thank you for your positive comments on the last version of our manuscript and for the opportunity to resubmit our work. We have taken on board the many useful comments offered during the last round of review and have revised our manuscript accordingly.

One of the outstanding concerns was how moths were treated in our phylogenetic analyses. In response to the nature of the reviewer concerns about the Lepidopteran data, we have altered how we deal with moths specifically and Lepidoptera generally in three important ways:

1. We confine our main analyses (comparing evolutionary rates between fertile and non-fertile sperm) to Lepidoptera, effectively combining moths and butterflies into a single analysis, as suggested by the reviewer.
2. We separate moths only for analyses that compare rates of evolution in sperm length and the female reproductive tract. Similar analyses across all Lepidoptera in our dataset were not possible as we did not have reproductive tract data for butterflies. We remain convinced that the manuscript benefits by the inclusion of moths in this particular analysis, since their inclusion demonstrates a repeated pattern of evolutionary responses in two distinct groups (moths and Drosophila) where the required data are available.
3. We stress in the Discussion and ESM the limitations of the Lepidoptera phylogeny used in our analyses, while also stressing that the pattern we observed (i.e. that one trait evolves faster than another) is robust, even if the estimates of the rates of evolution may change when using a different phylogeny – note that this is consistent with the view of the referee as well and maps onto one of the reviewer's suggestions for addressing this issue.

We feel that attending to these additional new referee comments has helped strengthen our manuscript and conclusions. We hope you agree.

Best wishes,

John Fitzpatrick
(on behalf of the other authors)

Associate Editor Board Member
Comments to Author:
Dear Authors,

I have now received two careful and constructive reviews for your manuscript "Repeated evidence that the accelerated evolution of sperm is associated with their fertilization function".

First, I would like to thank the authors for revising their manuscript and for the most part addressing all prior Reviewer concerns and comments. The restructured Introduction and Methods allow the reader to more easily and better link the motivation for the study with the subsequent analyses and data. I believe the manuscript is much improved and has the potential to be of really strong scientific importance and relevance for Proceedings B.

As with the previous submission, both Reviewers agree with me about the manuscript's potential. However, one Reviewer still has very relevant concerns and issues that need to be directly addressed: mainly regarding the phylogenetic estimations and analyses (for example the choice to still include moths). This issue partly came up in the last submission and was not directly addressed here.

It is my hope that these new comments again help in revising the manuscript and I look forward to reading the next submission.

Response: We thank the Associate Editorial Board Member for their positive comments on the last version of our manuscript. In the revised manuscript we deal with the outstanding concerns raised by the new third referee and detail our responses below. By attending to these additional new referee comments, we feel that our manuscript has been further strengthened.

Reviewer(s)' Comments to Author:

Referee: 2

Comments to the Author(s).

The authors have addressed all of the comments and concerns I raised in my review. My only suggestion, which they should feel free to ignore, is that they reconsider their new title and perhaps instead use their old title but with accelerated in place of rapid.

Response: We thank the referee for the suggestion. However, we have elected to keep our revised title, particularly in light of comments about the use of the word 'convergent' from the previous round of review.

Referee: 4

Comments to the Author(s).

This manuscript presents an analysis of sperm cell evolution and provides a test of the principal hypothesis for their extreme morphology diversity. The authors ask three questions:

- 1) Does sperm cell morphology evolve quicker than that of other male traits?
- 2) If so, does the higher rate of evolution in sperm relate to their function in fertilization?
- 3) Finally, do the various components of a sperm cell evolve in concert, or do some components evolve more quickly than others?

The authors recognize a naturally-occurring test of their first two questions in species groups

with heteromorphic sperm (i.e. species that produce fertile and non-fertile sperm morphs: moths, butterflies, and *Drosophila obscura*). This is a clever and original insight. By comparing morphological evolution in fertile and non-fertile sperm, which originate in the same tissue and occur in the same part of the body, the authors remove many confounding variables and isolate function as the primary predictor of evolutionary rate. The authors report faster rates of evolution in fertile sperm versus non-fertile sperm and wing length, and they find that rates of evolution in fertile sperm match those of the female reproductive tract, consistent with a role of fertilization in driving sperm evolution. The approach is innovative and the study has the potential to be impactful, but two serious issues should be addressed.

Response: We thank the reviewer for their kind assessment of our work and have addressed their issues and comments in detail below.

Major Comments

1. Unreliable phylogenies for Lepidopterans. The authors made a concerted effort to use the same or similar genetic sequences to estimate phylogenies in the three species groups (i.e. COII was used to construct and date trees for *Drosophila*, the related COI was used to construct and date trees for butterflies and, separately, moths), but this is a bit misguided, as the greater priority is to use sequences that allow reliable estimates of phylogenies over the timescales in question.

The three species groups used in this study represent different taxonomic ranks and have diversified over vastly different timescales – *Drosophila obscura* is a subgroup within a genus that arose ~15-20 my (Gao et al. 2007, Obbard et al. 2012), while Lepidoptera is an entire order of insects that arose ~190 my (Mitter et al. 2017) and butterflies are a superfamily within Lepidoptera that evolved ~80-140 my (Mitter et al. 2017). Mitochondrial DNA has a relatively high substitution rate, making it useful for estimating relationships among closely-related species. For the *Drosophila obscura* group, COII is likely a good choice, but COI is unreliable for estimating phylogenies among the Lepidopterans, which contain species from different families and superfamilies, due to saturation. The authors note that COI is the only gene available for the moths in their dataset. One potential option is to use published phylogenies to find the topology of species in the dataset and then fix this topology when building a tree in BEAST, thus using COI only to estimate divergence times. But even then, dates from the more deeply-diverged splits are unlikely to be accurate. A potential fix could be use 2nd and 3rd codon positions, which have lower substitution rates (see references within Obbard et al. 2012). An alternative route may be to drop the moths from the analyses.

It could be argued that this problem is unlikely to affect the key results. The phylogeny will be inaccurate, and the estimated rates of evolution will be inaccurate. But estimated rates will be inaccurate in the same way for all traits. It might not be wrong to conclude that one trait evolves faster than another.

Response: The referee raises a number of very good points here. We share the concerns about

the usefulness of applying mitochondrial genes to determine species relationships following more ancient divergence dates. Our aim is to be as transparent with the reader as possible about the strengths and limitations of our results. And clearly having a better resolved phylogeny would have improved our estimates.

However, as the referee points out, the potentially inaccurate divergence dates in our Lepidoptera phylogeny are unlikely to influence the key results and will be 'inaccurate' in the same way for all traits. Importantly we were careful in every version of the manuscript to not describe specific estimated rate values since these are subject to change depending on the phylogeny used in the analysis. Instead we described the relative difference in rates between traits, which are more robust to changes in phylogenies. We agree with the referee that even with the limits of our Lepidoptera phylogeny we can still conclude that one trait evolves faster than another. Therefore, the inferential insights offered by our work stand, despite the limitations in the phylogeny.

However, we don't discount the important issues raised by the referee and deal with these issues in the manuscript in a transparent way for the reader by doing three things, including:

1. We altered the text describing our tree building process (ESM Lines 30-60) to highlight that mitochondrial genes like COI can provide unreliable estimates among Lepidoptera, while also highlighting that these are the only sequences now available to build phylogenies for this group.
2. We present the results for moths, but include a new paragraph to the Discussion (lines 332-344) that focuses on the potential limitations of inferring evolutionary patterns from phylogenies using sub-optimal genes.
3. We are careful to not comment on the specific rate estimate values, as these are dependent on the phylogeny being used (see also our response to a comment below about the axis-labels for our figure).

2. A second significant issue is the splitting of Lepidopterans in the dataset into two insect groups: butterflies (a Lepidopteran clade) and moths (non-butterfly Lepidopterans). This appears to have been done because data on the female reproductive tract were only available for moths, but it creates the misleading impression that the evolutionary dynamics uncovered in this analysis (i.e. faster rates of evolution in fertile versus non-fertile sperm) are independently replicated three times in insects. Butterflies are nested within moths and did not evolve heteromorphic sperm independently, and their differential evolutionary rates should not be considered independent either. The valid approach would be to join moths and butterflies into a Lepidopteran tree for the majority of analyses, then prune out the butterflies for the test that includes the female reproductive tract. This change should be reflected in the reporting of the results.

Response: A good point. We have now treated Lepidopterans as a whole for our main analysis and only separated out the moths for the analyses assessing the female reproductive tract.

Minor Comments

Abstract:

1. Here and throughout the paper, the term “divergence” (e.g. phenotypic divergence, evolutionary divergence, morphological divergence) is used with reference to whole species groups, and to me this creates some confusion. When I think of divergence I think of the evolution of differences between two lineages, such as sister species that have recently diverged. I recommend replacing divergence with evolution to make it clearer that we’re talking about whole species groups.

Response: Done

2. Line 28-30. This sentence makes it sound as though the only component of sperm morphology that was studied was head length. This more interesting result that should be summarized here is that the two types of sperm have evolved quite differently (faster head evolution in fertile sperm, faster flagellum evolution in non-fertile sperm).

Response: Changed accordingly (lines 28-30).

3. Line 33. I think this would be stronger if the final sentence were about sperm, rather than the female reproductive tract. Faster evolution in the female reproductive tract was marginal and it was not the focus of the study.

Response: We have now deleted this sentence.

Introduction:

4. Line 64-67. This could use clarification. I think the point is that it’s difficult to draw conclusions from previous studies on the cause of different rates of evolution in sperm compared to other features. But there was nothing erroneous about drawing the conclusion that sperm evolved slower if that’s what the results showed. In general, I think the wording in this paragraph is a bit harsh on previous studies, which may have had different aims than the present paper. The emphasis should not be on why they were wrong but on how this study builds on them.

Response: Good point. We have now reworded parts of this paragraph to emphasize how our study builds on previous work (lines 51-72) and we revisit this point in the Discussion (see reviewer comment 16).

Methods (including Supplementary Methods):

5. The analytical framework is ideal for the question and appears to have been used correctly, though no code was provided for verification. It would be useful to have a column showing the

degrees of freedom for the likelihood ratio tests in Tables 1 and 2 and ESM Table 2, as these will be different for 2-trait “pairwise” trait models versus models with > 2 traits.

Response: Done, df have been added. We also now note in the supporting material that the R code to implement the analyses reported in the manuscript are already available in the Appendix of Adams 2013 (lines 141-142), which is why we did not include them here. However, if the Editor feels that the inclusion of this code would be helpful we are very happy to include it.

The authors might also include a correction for multiple likelihood ratio tests, which increases the chances of finding a significant p-value by chance (though this will be more of an issue for Table 2 than the key results in Table 1, which are far from marginal).

Response: We respectfully disagree. Corrections for multiple testing can be useful and necessary. But in cases where there is low statistical power to detect effects, due to small sample sizes, there is a greater probability in making a Type II error (not rejecting a null hypothesis when it is false) than making a Type I error (rejecting a null hypothesis when it is true). Correcting for multiple tests under these conditions is overly conservative and inflates Type II errors to unacceptable levels. These issues are nicely laid out by Shinichi Nakagawa’s work (Nakagawa 2004, A farewell to Bonferroni: the problems of low statistical power and publication bias. *Behavioral Ecology* 15:1044-1045). However, the suggestion by the same reviewer to add AIC values better contextualizes our results, so we added those (See Tables 1 and 2).

6. Readers would greatly benefit from a figure displaying the dated phylogenetic trees used in analyses – if not in the main paper, then at least in the supplementary materials. Important aspects of the trees are unclear from the text alone, such as the dates of key divergence events and whether the trees are fully resolved or if polytomies exist. Interested readers will certainly want to have a look at the estimated relationships among taxa, the dates of cladogenesis, and the patterns of branch lengths, all of which are easier to show than to describe verbally.

Response: The phylogenies were included in text format in the ESM in all previous submissions. However, we agree that visual presentation of the phylogenies is easier for readers to appreciate the relationships among taxa etc. and so now also include figures showing the dated phylogenies in the ESM (see Figures S1 and S2).

7. The authors derive age estimates from the software BEAST, which requires as input the mean and standard deviation of mutation rate in a given genetic sequence. Ideally, these parameters are based on previously published estimates of mutation rate for a locus derived from fossil or biogeographic calibration. The authors should clarify the value of these parameters used in the analysis (was it just default values?) and why they were chosen (i.e. what studies, if any, this estimate was based on).

Response: The parameter values were default BEAST values and we now state this in the text (ESM lines 79-80).

8. I suggest the authors drop the tests of phylogenetic signal. These tests were used to assess whether traits conform to the expectations of a Brownian motion process, which is an assumption of the models used in their main analyses. But phylogenetic signal is a poor indicator of the underlying evolutionary process (see Revell et al. 2008), and the authors already conduct the more appropriate check (likelihood-based model comparison, ESM Table 2). Also, the results of the phylogenetic signal tests don't appear to influence the traits that are eventually used in the main analysis (thorax length, which had significant phylogenetic signal, was excluded from downstream analysis, but seminal receptacle length was not).

Response: We thank the reviewer for suggesting this. Indeed, upon reflection, we agree. As the reviewer states, we had already conducted the more appropriate check for the assumption of Brownian motion and correspondingly dropped the poorly-performing trait from subsequent analyses.

9. Supp Mat: In the paragraph, "Drosophila and Lepidoptera Phylogenies", the phrase "monophyletic clade" is redundant. A clade is monophyletic by definition.

Response: Corrected.

10. Supp Mat: In the first paragraph under "Phylogenetic Analyses", the topic sentence is a bit misleading. The authors don't use a three step approach to determine whether traits conform to Brownian expectations; they use a two-step approach to determine if traits are Brownian, after which a subset of traits were used in the main evolutionary rate analysis.

Response: Corrected.

11. Supp Mat: In the last sentence of the first paragraph in "Phylogenetic Analyses", the wording suggests that data transformations were performed with the sole purpose of creating unitless variables, but I'd quickly note that an additional and very important purpose of the transformations is to bring the evolutionary change of different traits onto a common scale (i.e. to avoid the problem of comparing a 1mm change in sperm cell length to a 1mm change in wing length – changes that are proportionally very different).

Response: We have now added that data transformation allows different traits to be compared on a common scale (ESM lines 115-116).

12. Supp Mat: In the section "Assessing phylogenetic signal", "Blomberg's K" should be Blomberg's K.

Response: Yes, this was a typo. However, following the Referees earlier comment (point 8) we have deleted this entire section and no longer present the tests of phylogenetic signal.

13. Sup Mat: In the first paragraph of “Assessing phylogenetic signal”, the interpretation of Blomberg K values is incorrect. K values greater than 1 indicate that traits in different lineages have diverged less (i.e. are more similar) than expected under BM.

Response: Thank you for pointing this out. However, as noted above, this section has now been deleted.

Results

14. Some readers will be expecting to see AICc values (or, alternatively, delta AICc values) reported in tables 1 and 2, along with the results of likelihood ratio tests, as the former are more commonly encountered. Presenting the results this way should have no effect on the paper’s conclusions but may help some readers assess confidence in model comparisons.

Response: Done. We note that this information was in the original submission of our manuscript and was requested to be removed by the referees following the first review. We are happy to return this information to the text as we agree with the referee that the inclusion of AIC values increases confidence in the model comparisons.

15. The y-axes in the three panels in figure 1 should be equivalent.

Response: We respectfully disagree. Estimates of rates of evolution are dependent on the phylogeny being used in the analysis. Since the *Drosophila* and *Lepidoptera* analyses were performed on separate phylogenies it would be misleading to present the panels of Figure 1 with equivalent y-axes. In fact, we would like to actively discourage readers from attempting to compare directly the evolutionary rates values from one panel to another as these values are not comparable. We have now altered the figure caption to state this explicitly.

Discussion:

16. In the discussion, I’d like to see the authors refer back to the studies in mentioned in line 59. It would be interesting to hear why the results appear to be different in these insects than in beetles and *Anolis* lizards.

Response: We now address this in the first paragraph of the Discussion (lines 241-245) and suggest that the big difference between our study and previous studies is that previous approaches compared rates of sperm length evolution with rates of traits that operate in other selective environments and therefore face different constraints.

Refs

Obbard, DJ. et al. 2012. Estimating divergence dates and substitution rates in the *Drosophila* phylogeny. *Molecular Biology and Evolution* 29: 3459–3473.

Gao et al. 2007. Molecular phylogeny of the *Drosophila obscura* species group, with emphasis on the Old World species. *BMC Evolutionary Biology* 87:

Mitter, C. et al. 2017. Phylogeny and evolution of Lepidoptera. *Annu. Rev. Entomol.* 62: 265–83.

Revell, L.J. et al. 2008. Phylogenetic signal, evolutionary process, and rate. *Systematic Biology*.

Appendix C

Author responses (in blue text):

Associate Editor Board Member

Comments to Author:

We have now received back comments from a statistical reviewer for the manuscript "Repeated evidence that the accelerated evolution of sperm is associated with their fertilization function". I think the main positive takeaway is that the reviewer feels the overall main results of the paper, especially as they pertain to the *Drosophila* phylogeny, are compelling. I wholeheartedly agree. However, I have shared their and others concern that the Lepidopteran results are less strong, and instead take away from the overall scope of the manuscript as written (or analyzed). There have now been a few attempts to fix these issues but, as the Reviewer nicely points out in their comments, glaring ones still remain. If the authors are not able to directly address the comments of the Reviewer they may find more success sending their paper elsewhere. If they are able to address these comments we would more than welcome the resubmission.

Response: We have now directly addressed the main issue regarding the Lepidoptera phylogeny. Specifically, we now use the recently published genomic and transcriptomic-derived Lepidoptera phylogeny suggested by the reviewer to compare rates of evolution among traits. Using this robust phylogeny, we find that fertile sperm evolve faster than non-fertile sperm in Lepidoptera. Importantly, this result is consistent with the findings reported in previous versions of our manuscript. However, by following the suggestions for the reviewer, we are now able to demonstrate this difference in evolutionary rates between sperm types in Lepidoptera using a far more robust phylogeny.

Reviewer(s)' Comments to Author:

Referee: 4

Comments to the Author(s).

I sincerely thank the authors for their time and effort in addressing several of the issues identified in the previous manuscript. In most respects I am satisfied with the paper in its current form. However, the Lepidopteran phylogeny stands out as an issue.

Response: We thank the referee for their kind comments and continued attention to our manuscript. The referee has clearly put a lot of thought into their review and we appreciate it.

While I recognize the limitations to sequence data available in genbank and empathize in this regard, the problem remains that COII sequences probably evolve too quickly to accurately resolve relationships among the Lepidopterans included in this study. If this tree had been constructed to simply "correct" for phylogeny given some sort of regression analysis, this would

be less of an issue, as several studies have shown a compromised phylogeny to be preferable to no phylogeny at all. However, in the present case, the primary results of the paper are based on estimates of evolutionary rates that are directly derived from estimates of branch lengths in the phylogeny. Confidence in evolutionary rate estimates are thus limited by low confidence in branch length estimates derived from COII. Presented as is, I think readers would view these Lepidopteran results with strong suspicion. I have three suggestions the authors may consider.

1. Re-run the Lepidopteran analysis using the most comprehensive Lepidopteran tree available in the literature.

- Based on my very quick search, this would be the Kawahara et al. (2019 PNAS) tree, which is based on a huge genomic and transcriptomic dataset. This tree contains 31 of the genera used in the present analysis, so the Lepidopteran test could easily be re-run in a short period of time using this published tree (pruned to the 31 relevant tips). The authors could decide to restrict their analysis to just these 31 lineages, or, if such an analysis were to give them very similar results to what they report here, they could use it to say that their larger results are unaffected by using just COII.

Response: We agree that using a more robust phylogeny is the best way to move forward with confidence in our findings and we thank the referee for this constructive suggestion. The phylogeny recently published by Kawahara et al. (2019 PNAS) does indeed appear to be the best available phylogeny for a broad range of Lepidoptera currently available. We use that phylogeny in the analyses in our revised manuscript.

The referee correctly points out that many of the genera of the species in our Lepidoptera dataset are present in the phylogeny. However, rather than use species placed based on their genus, we first restricted our analysis to species that were present in Kawahara et al.'s phylogeny. This approach matches with the species-level placement in the *Drosophila* species we examine in the first part of our manuscript. Specifically, 12 of the Lepidopteran species in our dataset were present in the Kawahara phylogeny, allowing us to test one of our main findings (i.e. fertile sperm evolve faster than non-fertile sperm) on a more robust phylogeny, albeit with a restricted sample size. However, we were not able to assess the other main finding in Lepidoptera (i.e. similar rates of evolution in fertile sperm and female reproductive organs) as only one species in this restricted dataset had data available on spermathecal duct length.

When we pruned the tree to only include these 12 species and compared evolutionary rates among fertile sperm, non-fertile sperm, and wing length in Lepidoptera we found fertile sperm evolve faster than non-fertile sperm in Lepidoptera. Moreover, the pattern of evolutionary rates among traits remained consistent with the results reported in our earlier submission – fertile sperm evolves faster, wing length evolve at an intermediate rate and non-fertile sperm evolves slowest. While the pattern is consistent the reduced sample size likely reduced the power to adequately detect differences between wing length and sperm type.

We further explored the reviewer's suggested approach by expanding our analysis to fit species based on whether their genera were present in the phylogeny. This analysis revealed very

similar results to those observed when we assess the 12 species that were present in Kawahara's phylogeny. Therefore, we only present the results from the species level analysis in our revised manuscript.

We restrict all analyses presented in the main text to those using the Kawahara et al. tree when assessing Lepidoptera. However, since this dramatically reduces the sample size in our analysis – from 135 to 12 – we also include an analysis comparing fertile and non-fertile rates of evolution using the COI-derived tree in the supplemental material. This additional analysis backs up the analyses presented in the main text. However, we were careful to clearly state the caveats when using a COI-derived tree in the supplemental material. In addition, we intentionally refrained from including female reproductive organ data in the analyses presented in the supplemental material as we were not able to assess this trait in the results presented in the main text.

2. Remove Lepidoptera from the analysis.

- In my opinion, the results from *Drosophila* alone are sufficient to substantiate each of the key findings in this paper (i.e. faster rates of evolution in fertile versus non-fertile sperm, matching rates of evolution in fertile sperm and female reproductive organs, and differential rates of evolution in different sperm components), all of which are interesting. The manuscript would be relatively unchanged by focusing on just *Drosophila* (with the exception of the title).

Response: We agree with the reviewer and editorial board member that the *Drosophila* results alone are compelling. However, we argue that the impact of the paper is substantially increased by the inclusion of Lepidoptera, a second group with sperm heteromorphism. We have therefore elected to keep Lepidoptera in our revised manuscript, while taking on board the referee's helpful suggestions about how best to do this.

3. Demonstrate that the issue with COII is unimportant.

- The authors could provide a simple saturation plot to demonstrate that the COII sequence data has sufficient "depth" to estimate the rates reported here. If the plot shows little or no saturation, then the tree and the rate estimates would be substantiated.

Response: We find that the pattern of the main result of fertile sperm evolving faster than non-fertile sperm is unaffected by which phylogeny we use. However, because we were not able to test female reproductive morphology in Lepidoptera due to reduced sample size using the more robust Kawahara phylogeny, we do not present the results using the COI derived phylogeny in the main text.

I think that none of the above suggestions would dramatically alter the current paper or take much time, but they would alleviate the main methodological concern that readers would probably have from the Lepidopteran results in their current form.

Response: We hope that by following the referee's suggestion that their main methodological concern is now addressed in full.

Minor comments:

1. Line 54-55: wording here is confusing. Would suggest something shorter along the lines of "however, in these studies, rates of sperm diversification were compared ..."

Response: Good suggestion. This has been changed accordingly (line 55).

2. Line 260: mixed-up wording. the "and" should be moved from its position to go between "organ length" and "sperm length"

Response: Corrected.

3. I would prefer to see the paragraph on phylogenetic tree construction in the results section itself, rather than in supplementary methods. It doesn't seem particularly long and readers could more directly assess confidence in results without having to dig into online supplements.

Response: We have added the description of the phylogenetic construction to the main text to give the reader the chance to directly assess the confidence of our results based on the phylogeny used (lines 158-199).

4. Wording regarding the main test of the paper is confusing in two areas.

Response: We trust the changes we have made, which do not change the main test of the paper, resolve this comment.

Appendix D

Dear Professor Carvalho and Associate Editor Board Member,

Thank you for considering our manuscript once again. We have now made the minor changes suggested by the referee and editorial board member to specify the limitations of the scope of the Lepidopteran analysis in the abstract and discussion.

Best wishes,
John Fitzpatrick
(on behalf of the other authors)

Author responses (in blue text):

Associate Editor Board Member
Comments to Author:
Dear Authors,

Your paper has now been reassessed by a past reviewer and myself. Both of us find the manuscript much improved once again, and commend the authors on a fine job of answering all the reviewer queries and comments. The reviewer has one final important comment that should be addressed by the authors. Namely, they would like the authors to specifically address the obvious limitations in scope for the Lepidopteran data. I agree that this is a much needed step, and one that will not take away from the main and novel inferences that has been made from these data sets. I look forward to the revised manuscript.

Response: We have now highlighted the limitations of the scope of the Lepidopteran analysis in the abstract and discussion as suggested.

Reviewer(s)' Comments to Author:

Referee: 4

Comments to the Author(s).

I thank the authors for their attention to previous concerns regarding the Lepidopteran phylogeny.

The main issue with the previous manuscript was that the authors inferred evolutionary rates for sperm traits based on a Lepidopteran phylogeny constructed from COII, a mitochondrial gene subunit with a substitution rate that is probably too high to accurately infer relationships among such anciently-diverged lineages. I had no major criticisms of the *Drosophila* results, which were and are sufficient to substantiate the paper's main findings.

In the revised manuscript the authors employ a recently-published Lepidopteran phylogeny based on extensive genomic and transcriptomic datasets to estimate rates of evolution in sperm traits. Branch lengths in this phylogeny are much more reliable than the previous tree

based on COII, and as a consequence the estimates of evolutionary rate are also much more reliable. The cost to this improved approach is that only 12 of the 135 Lepidopteran species for which the authors obtained trait data could be included in the revised analysis. Nevertheless, the analysis found quite strong support for a more complex model of trait evolution (i.e. a model in which different traits had different evolutionary rates). I am therefore satisfied with these newer results.

One final and minor issue concerns the scope of inference for Lepidoptera. I think the authors need to be clearer in the abstract and in the discussion that their results, while interesting and novel, are true for *Drosophila obscura* and a small group of Lepidopterans for which data were available, not necessarily for Lepidoptera as a whole. For instance, the authors write "We provide consistent evidence that fertile sperm length evolves faster than non-fertile sperm length in *Drosophila* from the *obscura* group and Lepidoptera, two groups with heteromorphic sperm." To me the Lepidopteran part of this sentence is going a bit far, as the inference is based on 12 species and Lepidoptera contains ~180k described species. A slight wording change to something along the lines of "We provide consistent evidence that fertile sperm length evolves faster than non-fertile sperm length in *Drosophila* from the *obscura* group and in a group of Lepidopterans for which sperm trait data were available ..." would clear that up and leave me no further objections.

Response: We now specify in the Abstract that we are assessing 'a subset of Lepidoptera species' (line 21) and in the Discussion we have made the changes suggested by the referee (lines 254-255) as well as pointing out the limited sample size more explicitly in line 271.